# Single-cell transcriptomics of vomeronasal neuroepithelium reveals a differential endoplasmic reticulum environment amongst neuronal subtypes

GVS Devakinandan[1]*, Mark Terasaki[2], Adish Dani[1]*

[1]Tata Institute of Fundamental Research, Hyderabad, India; [2]Department of Cell Biology, University of Connecticut Health Center, Farmington, United States

**\*For correspondence:**
devakinandangvs@gmail.com
(GVSD);
adishd@tifrh.res.in (AD)

**Competing interest:** The authors declare that no competing interests exist.

## eLife Assessment

This is a **valuable** manuscript analyzing single-cell RNA-sequencing data from the mouse vomeronasal organ. **Convincing** evidence in this manuscript allows the authors to identify and verify the differential expression of genes that distinguish apical and basal vomeronasal neurons. The authors also show that Gnao1 neurons exhibit enriched expression of ER-related genes, which they verify with in situ hybridizations and immunostaining and also explore via electron microscopy.

**Abstract** Specialized chemosensory signals elicit innate social behaviors in individuals of several vertebrate species, a process that is mediated via the accessory olfactory system (AOS). The AOS comprising the peripheral sensory vomeronasal organ has evolved elaborate molecular and cellular mechanisms to detect chemo signals. To gain insight into the cell types, developmental gene expression patterns, and functional differences amongst neurons, we performed single-cell transcriptomics of the mouse vomeronasal sensory epithelium. Our analysis reveals diverse cell types with gene expression patterns specific to each, which we made available as a searchable web resource accessed from https://www.scvnoexplorer.com. Pseudo-time developmental analysis indicates that neurons originating from common progenitors diverge in their gene expression during maturation with transient and persistent transcription factor expression at critical branch points. Comparative analysis across two of the major neuronal subtypes that express divergent GPCR families and the G-protein subunits Gnai2 or Gnao1, reveals significantly higher expression of endoplasmic reticulum (ER) associated genes within Gnao1 neurons. In addition**,** differences in ER content and prevalence of cubic membrane ER ultrastructure revealed by electron microscopy, indicate fundamental differences in ER function.

## Introduction

The Vomeronasal organ (VNO), part of the vertebrate accessory olfactory system, is a major pheromone-sensing organ, that is thought to be specialized to evoke innate social behaviors in mammals. The rodent VNO neuroepithelium consists of two major neuronal subtypes, that are defined primarily based on the expression of two distinct families of G-protein coupled receptors (GPCRs) and their associated G protein alpha subunit: vomeronasal type-I GPCRs (V1R) with Gαi2 subunit (Gnai2) neurons or vomeronasal type-2 GPCRs (V2R) with Gαo subunit (Gnao1) neurons. V1R and V2R expressing neurons are spatially segregated within the neuroepithelium, and project to distinct locations in the accessory olfactory bulb along the anterior-posterior axis. Functional evidence suggests that V1R and V2R

neurons may recognize ligands that mediate different behaviors (*Chamero et al., 2011*; *Oboti et al., 2014*; *Trouillet et al., 2019*; *Pallé et al., 2020*). However, we currently lack a comprehensive mapping of receptor-ligand interactions and how these lead to distinct behaviors. Cognate ligand binding in both Gnao1 and Gnai2 neurons, is thought to signal via the phospholipase-c pathway which results in the activation of the VNO-specific cation channel, Trpc2 (*Liman et al., 1999*; *Stowers et al., 2002*; *Zufall, 2005*).

In the main olfactory epithelium (MOE), neurons regenerate throughout life, continuously differentiating from a stem cell population into neurons and supporting cell types (*Fletcher et al., 2017*). Neuronal differentiation is also associated with the expression of a single GPCR to the exclusion of many others (*Hanchate et al., 2015*; *Tan et al., 2015*), leading to the hallmark feature termed as 'one neuron one receptor (ONOR)' rule (*Malnic et al., 1999*; *Serizawa et al., 2000*; *Serizawa et al., 2003*). VNO sensory neurons (VSNs) also continuously regenerate (*Barber and Raisman, 1978*; *Wilson and Raisman, 1980*) from a stem cell population, and differentiating cells take on the identity of supporting cells and mature neurons of either the Gnao1 or Gnai2 sub-types (*Katreddi et al., 2022*). While Gnai2 neurons seem to largely follow the ONOR rule in expressing one V1R gene per neuron (*Serizawa et al., 2004*), Gnao1 neurons seemingly deviate from this rule by co-expressing a combination of GPCRs. Thus, each Gnao1 neuron expresses a single V2R gene from the ABD family as well as a member of the C family V2R genes (*Martini et al., 2001*; *Silvotti et al., 2007*; *Ishii and Mombaerts, 2011*). In addition, further molecular diversification of Gnao1 neurons occurs due to combinatorial co-expression of a family of non-classical Major Histocompatibility Complex (MHC) genes, H2-Mv, that are co-expressed along with V2Rs (*Ishii et al., 2003*; *Loconto et al., 2003*, *Ishii and Mombaerts, 2008*). A few combinations of V2R and H2-Mv expression have been identified, indicating that these co-expression patterns are non-random and could be functionally important (*Martini et al., 2001*; *Ishii et al., 2003*; *Loconto et al., 2003*; *Silvotti et al., 2007*; *Ishii and Mombaerts, 2008*; *Ishii and Mombaerts, 2011*). However, identifying the precise combinatorial expressions of V2R-ABD, V2R-C and H2-Mv has been elusive, partly due to the challenges in generating RNA or antibody reagents to identify closely homologous genes, combined with the challenges with multiplexing more than three targets.

Starting with a common progenitor population, how do cells of the VNO sensory epithelium diversify in their gene expression patterns that define the sensory neuron types? Recent studies using single-cell transcriptomic analysis have implicated Notch signaling along with downstream transcription factors, *Tfap2e*, *Bcl11b* to play a role in specifying the fates of developing VSNs (*Enomoto et al., 2011*; *Katreddi et al., 2022*; *Lin et al., 2022*). As the progenitor cells differentiate to mature neurons, transcriptional reprogramming downstream to Notch signaling leads to distinct neuronal populations: Gnao1 and Gnai2 neurons. While these studies identified transcription factors, it is not clear whether the two major differentiated VSN subtypes expressing Gnao1-V2Rs or Gnai2-V1Rs fundamentally differ in their cellular properties.

Here, we applied a multilevel transcriptomics approach which bundles single-cell RNA sequencing (scRNA seq) validated by low-level spatial transcriptomics to characterize gene expression of the mouse VNO sensory epithelium. Our results identified genes specifically expressed in sensory, and non-sensory cell types and the divergence of gene expression between Gnao1 - Gnai2 neurons at different maturation stages. Analysis of mature neurons provided a comprehensive picture of specific co-expression patterns of VR and H2-Mv genes. Surprisingly, mature Gnao1 neurons revealed a significant enrichment of endoplasmic reticulum-related gene transcripts and proteins, which is indicative of a fundamental divergence in cellular function between these cell types. Electron microscopy further revealed a hypertrophic ER ultrastructure consisting of gyroid cubic membrane morphology, more prevalent in Gnao1 neurons.

## Results
### Single-cell transcriptome of mouse vomeronasal neuroepithelium
We standardized VNO neuroepithelium dissociation to obtain a single-cell suspension along with optimizing cell viability. We performed single-cell droplet-based capture coupled with scRNA seq, using the 10 X genomics platform on four biological replicates consisting of male and female animals. Initial quality control steps were performed (see methods) to remove cells with high or low total RNA

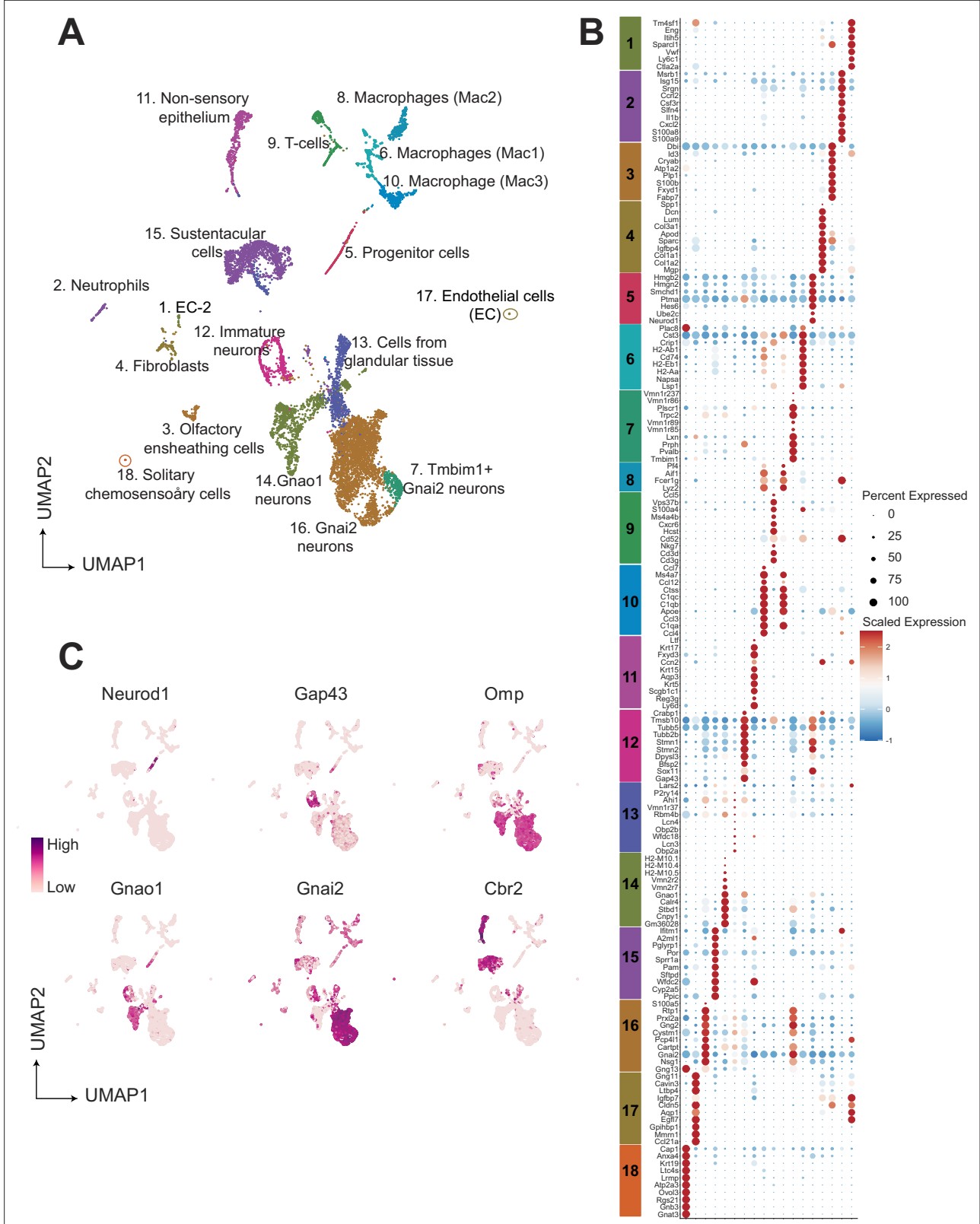

**Figure 1.** Single-cell RNA sequencing of the mouse vomeronasal neuroepithelium. (**A**) Uniform Manifold Approximation and Projection (UMAP) of 9185 cells from vomeronasal neuroepithelium with 18 identified clusters. Each point represents a cell that is color-coded according to the cell type. Clusters were assigned a cell type identity based on previously known markers. Percentage of total cells corresponding to each cluster is in *Supplementary file 2*. (**B**) Dot plot showing the expression of the top 10 gene markers for each cluster identified by differential expression

*Figure 1 continued on next page*

*Figure 1 continued*

analysis. Gene expression markers shared across multiple clusters are listed only once. The dot radius and dot color indicate the percentage of cells expressing the gene and scaled expression value in that cluster, respectively. Top 20 gene markers for each cluster with log2 fold change are listed in *Supplementary file 1*. (**C**) Feature plot showing the expression level of major known gene markers of neuronal and sustentacular cell types. *Figure 1—figure supplement 1* shows UMAP clusters and gene expression comparison between males and females.

The online version of this article includes the following source data and figure supplement(s) for figure 1:

**Figure supplement 1.** Comparison of cell type composition and gene expression from the male and female vomeronasal neuroepithelium.**Figure supplement 1—source data 1.** Average log2 fold change gene expression between male and female along with the fraction of cells expressing the gene in male or female cluster.

content and red blood cell contamination, resulting in the retention of 9185 cells from all samples for further analysis. Since our comparison of gene expression obtained from male versus female mice, did not reveal appreciable differences other than known sexually dimorphic genes that include *Eif2s3y*, *Ddx3y*, *Uty,* and *Kdm5d* (*Figure 1—figure supplement 1*), we pooled cells from both sexes for further analysis.

To group cells based on their gene expression profiles, we performed dimensionality reduction and unsupervised clustering, which resulted in 18 distinct clusters (*Figure 1A*), with gene expression markers specific to each cluster (*Figure 1B*). Major markers for each cluster were identified by differential expression analysis and the cell type identity was assigned to each cluster based on known markers for each cell type. Amongst 9185 cells, we observed that 63.7 percent (5852 cells) were neuronal cell types at different developmental stages which includes progenitor cells (Cluster 5; defined by

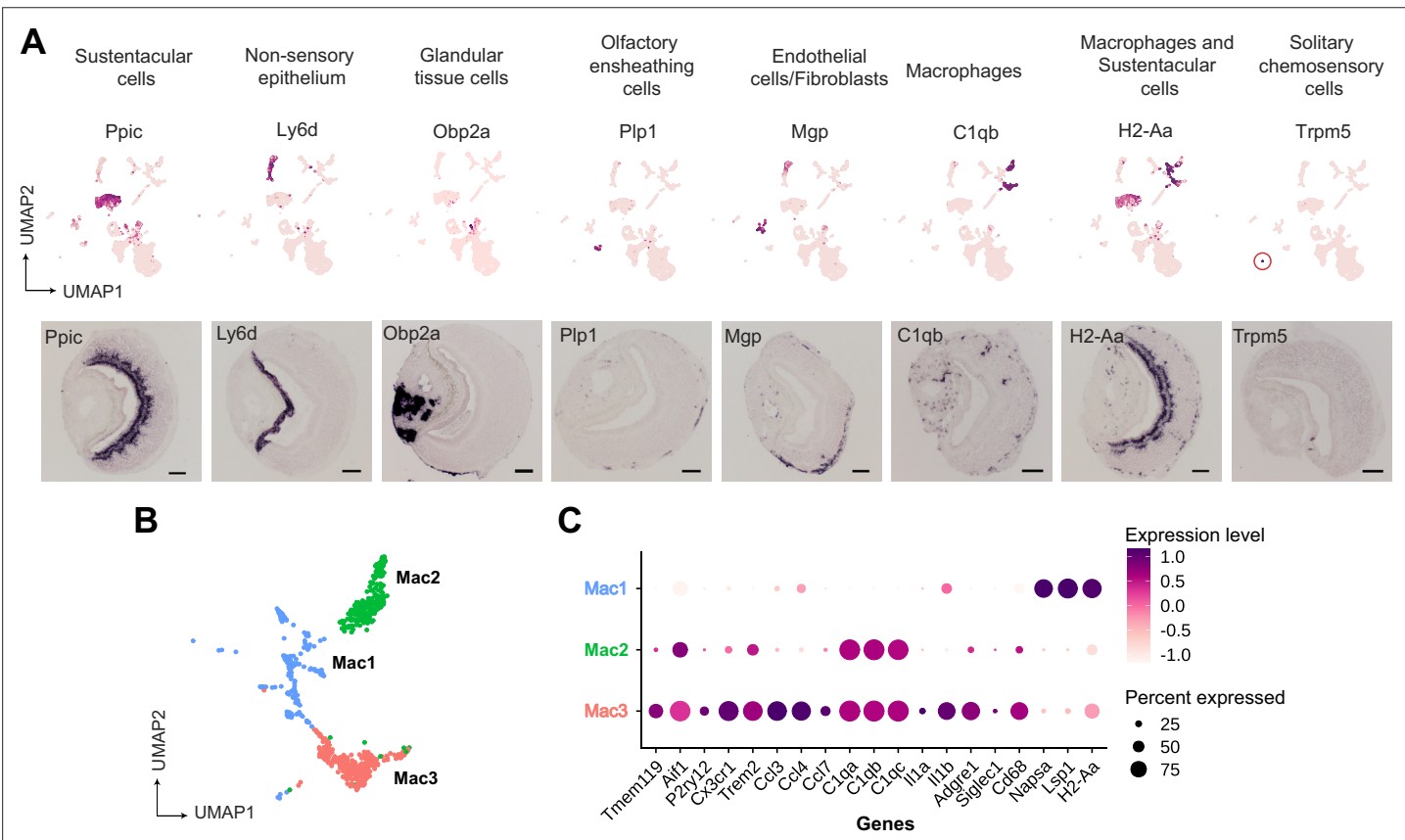

**Figure 2.** Gene markers of non-sensory cells in the vomeronasal organ identified from single-cell RNA sequencing. (**A**) Uniform Manifold Approximation and Projection (UMAP) feature plot (top) of a single representative gene marker for non-sensory cell clusters and their spatial gene expression pattern by RNA in situ hybridization on Vomeronasal organ (VNO) coronal sections (bottom, scale bar: 100 μm). (**B, C**) Three clusters of macrophage-like cell types represented by UMAP projection (**B**) and a dot plot (**C**) showing differentially expressed genes between these clusters. The dot radius indicates the percentage of cells expressing a gene, whose scaled expression level is represented by the intensity of color as per the scale. Additional markers of each non-sensory cell type are in *Supplementary file 1* and differential expression between macrophage clusters is in *Supplementary file 3*.

expression of *Ascl1*, *Neurod1*, *Neurog1*), immature neurons (Cluster12; expression of *Gap43*), Gnao1 neurons (Cluster 14; based on expression of *Gnao1*, V2Rs), and Gnai2 neurons (Cluster 7, 16; characterized by *Gnai2*, V1Rs expression) (*Figure 1C*). The remaining 36.3 percent of cells were classified to be of non-neuronal types which include sustentacular cells (Cluster 15; defined by *Cyp2a5*, *Ppic*), cells from non-sensory epithelium (Cluster 11; defined by *Ly6d*, *Krt5*), cells from glandular tissue (Cluster 13; characterized by *Obp2a*, *Lcn3*), immune cells which include macrophages (Cluster 8, 6, 10; defined by *C1qa*, *C1qb*, *H2-Aa*), T cells (Cluster 9; defined by *Cd3d*, *Cd3g*), neutrophils (Cluster 2; defined by *s100a8*, *s100a9*), olfactory ensheathing cells (Cluster 3; defined by *Gfap*, *Plp1*, *Sox10*), endothelial cells (cluster 1 and 17; defined by *Gng11*, *Egfl7*, *Eng*), fibroblasts (Cluster 4; defined by *Igfbp4*, *Dcn*, *Lum*) and solitary chemosensory cells (Cluster 18; defined by *Trpm5*, *Gnat3*). A complete list of the top 20 differentially expressed genes associated with each cluster and proportion of cells are provided in *Supplementary file 1* and *Supplementary file 2*.

We performed RNA in situ hybridization (RNA-ISH) on VNO sections to confirm the spatial classification of various non-sensory cell type clusters based on some of the genes identified in our study (*Figure 2A*). Thus, we confirmed *Ppic* to be expressed in sustentacular cells and *Ly6d* in cells of non-sensory epithelium bordering the VNO lumen. Since, *Cbr2* - a known marker of sustentacular cells is also expressed in the non-sensory epithelium (*Figure 1C*), *Ppic* and *Ly6d* are better markers to distinguish them. *Obp2a* was confirmed to be expressed in cells from glandular tissue on the non-sensory side. Plp1 and Mgp, markers of olfactory ensheathing glial cells and endothelial cells/fibroblasts, respectively, were observed to be expressed in cells located at the periphery of the neuro-epithelium. Macrophage cell markers *C1qb* and *H2-Aa* were observed to be specifically expressed in scattered cells throughout the neuroepithelium, with *H2-Aa* co-expression also seen in sustentacular cells (*Figure 2A*), confirming the presence of these cell types within the neuroepithelium.

Since our study identified multiple immune cell clusters, we examined these in some detail. Three clusters (*Figure 1A*, clusters 6, 8, 10 termed as *Mac1*, *Mac2*, *Mac3*, respectively) comprised of macrophage-like cells based on the expression of *C1qa*, *C1qb*, *C1qc*, and MHC class II gene - *H2-Aa* (*Figure 2B and C*). Further analysis indicates that cluster 10 (Mac3) expresses classical microglial markers such as *Tmem119*, *Aif1* (Iba1), *P2ry12*, *Cx3cr1*, *Trem2*, along with chemokine genes - *Ccl3*, *Ccl4*, and *Ccl7* and Interleukins – *Il1a*, *Il1b* (*Figure 2C*), indicating an activated macrophage subset. Cluster 8 (Mac2) is similar to Mac3 in regard to expression of classical tissue-resident macrophage marker - *Adgre1* (F4/80), *C1qa*, *C1qb*, *C1qc* but lacks expression of chemokine and interleukin genes mentioned earlier. In comparison to Mac3 and Mac2, the Mac 1 cluster is distinguished by the lack of *C1qa*, *C1qb*, *Cd68*, *Adgre1*(F4/80), and microglial markers but possesses high levels of *Napsa*, *Lsp1*, *H2-Aa* (*Figure 2C*, *Supplementary file 1* and *Supplementary file 3*). In summary, the identification of discrete macrophage cell types from our data, along with experimental validation of their presence using some of the expression markers, points to the possibility of tissue-resident or tissue-infiltrating macrophages within the VNO neuroepithelium. These VNO macrophage types could have functional implications toward influencing processes such as neuronal differentiation or responses to tissue repair and injury.

On the other hand, we did not observe evidence via RNA-ISH for T-cell markers (*Cd3d*, *Cd3g*), raising the possibility that these immune cell types may have co-purified from blood. Lastly, we were able to identify via RNA-ISH, the expression of *Trpm5* (*Figure 2A*) and *Gnat3* genes from our dataset, that mark solitary chemosensory cells expressing taste receptors and related markers (*Ogura et al., 2010*; *Supplementary file 1*), indicating that rare cell-populations were captured, and their markers identified in our study.

## Neuronal cell types and transient gene expression in developing neurons

Our next step was to perform in-depth analysis of vomeronasal sensory neurons. In order to obtain a better resolution on neuronal sub-types, We pooled cells from clusters representing neuronal types identified in *Figure 1* (clusters 5, 7, 12, 14, 16), created a new Seurat object, and performed re-clustering. This resulted in 13 clusters (labeled n1-n13), which potentially represent distinct neuronal sub-populations within the broadly defined Gnao1 and Gnai2 neurons (*Figure 3—figure supplement 1A*). Since neurogenesis is a continuous process in VNO neurons (*Barber and Raisman, 1978*; *Wilson and Raisman, 1980*), these clusters also represent a snapshot of cells at different developmental states,

identified by known markers: Globose basal cells (Cluster n5; *Ascl1*), progenitor cells (Cluster n5; defined by expression of *Neurod1*, *Neurog1*), immature neurons (Cluster n6, n7; defined by *Gap43*) expressing *Gnao1* or *Gnai2* (*Figure 3—figure supplement 1B*). Among Gap43 negative mature neurons, Gnao1 neurons are clusters n1-n4, (*Figure 3—figure supplement 1A, B*, corresponding to cluster 14 in *Figure 1A*), where sub-cluster identity is clearly defined by their co-expression of H2-Mv and Vmn2r GPCRs (discussed in detail in the co-expression section below). In contrast, six clusters (n8-n13) correspond to mature Gnai2 neurons (clusters 7, 16 from *Figure 1A*), of which n8-n11 do not have any definite gene markers and differ based on differences in the expression level of a few genes (*Figure 3—figure supplement 1C*, *Supplementary file 4*). Neurons of sub-cluster n12 (cluster 7 from *Figure 1A*) exclusively express genes *Tmbim1*, *Lgi2*, *Cleg2g,* and n13 are distinguished by the expression of previously identified activity-dependent genes - *Rasgrp4*, *Pcdh10,* and *S100a5* (*Fischl et al., 2014*; *Figure 3—figure supplement 1C*, *Supplementary file 4*). To our knowledge, unlike mature Gnao1 neurons, mature Gnai2 subclusters n8-n11 and n13, do not have an obvious principal parameter for their subclassification.

Expression of cognate V1R or V2R GPCRs is one of the hallmarks of VSN differentiation. We asked the question, to what extent do VRs influence neuronal sub clustering. Exclusion of VRs from clustering parameters did not affect the divergence of mature neurons into Gnai2 /Gnao1 and only selectively affected some of the subclusters (*Figure 3—figure supplement 2*), thus indicating that neuronal subtype identity was governed by gene expression differences beyond vomeronasal GPCRs.

Since immature neurons are separated into two clusters expressing Gnao1 or Gnai2, these imply a possible branch point in the developmental trajectory towards mature neurons. To confirm this, we performed pseudo-time analysis on neurons using Slingshot, and progenitor neurons (cluster n5) as a start/anchor point. For the purpose of developmental analysis, we merged subclusters within mature Gnao1 and Gnai2 neurons. The trajectory of development confirms a split of immature *Gap43* +neurons into Gnao1 or Gnai2 cell clusters (*Figure 3A*). We performed differential expression analysis between immature neurons (cluster n6: *Gap43+*, *Gnao1* + vs cluster n7: *Gap43+*, *Gnai2*+) (*Figure 3B*). To identify potential transcription factors in this developmental branch point, we manually curated genes that have molecular features associated with DNA binding domains or transcription regulation based on literature (*Figure 3C*). To our surprise, we found multiple genes that are known transcription regulators but are not previously reported to be enriched in Gnao1 neurons. These include *Creb5*, *Prrxl1*, *Shisa8*, *Lmo4,* and *Foxo1*. *Creb5*, *Prrxl1* are transcription factors expressed only in immature Gnao1 neurons whereas *Shisa8*, *Lmo4,* and *Foxo1* expression is enriched in immature Gnao1 neurons compared to mature ones (*Figure 3C*). *Prrxl1* was reported to auto-repress its expression which explains how its expression is limited to immature Gnao1 neurons (*Monteiro et al., 2014*). These data indicate that the developmental transition of VNO neurons to Gnao1 lineage may involve the transient expression of transcription factors within immature neurons after dichotomy is established.

## Co-expression of vomeronasal receptors

One of the features that differentiate vomeronasal sensory neurons from main olfactory sensory neurons is their systematic deviation from the 'one neuron one receptor rule,' especially in mature Gnao1 neurons. V2R receptors have been classified into families-A, B, D, E, and a distinct family-C that is phylogenetically closer to calcium-sensing receptors (*Young and Trask, 2007*). Several studies have established a broad pattern of co-expression amongst these V2R family members, such as members of family ABDE co-express with family-C GPCRs and components of family-C (*Vmn2r1* -also termed -C1, *Vmn2r2-Vmn2r7* -termed C2) in turn co-express with each other in specific combinations (*Martini et al., 2001*; *Silvotti et al., 2007*). Furthermore, H2-Mv genes within Gnao1 neurons also exhibit non-random co-expression patterns with the V2R GPCRs. (*Ishii et al., 2003*; *Loconto et al., 2003*; *Silvotti et al., 2007*; *Silvotti et al., 2011*; *Ishii and Mombaerts, 2008*; *Ishii and Mombaerts, 2011*). To further investigate these co-expression patterns and subpopulations within Gnao1 neurons, we took a closer look at clusters n1-n4 comprised of mature Gnao1 neurons (*Figure 3—figure supplement 1A*). These four clusters (*Figure 4A*) are found to be organized majorly based on the expression of family-C V2Rs and H2-Mv genes, especially Vmn2r1 or Vmn2r2 (*Figure 4B–E*). Of the four clusters, cluster n1 and cluster n4 express Vmn2r1 (*Figure 4B and E*) whereas cluster n3 neurons are distinguished by the co-expression of *Vmn2r2* along with

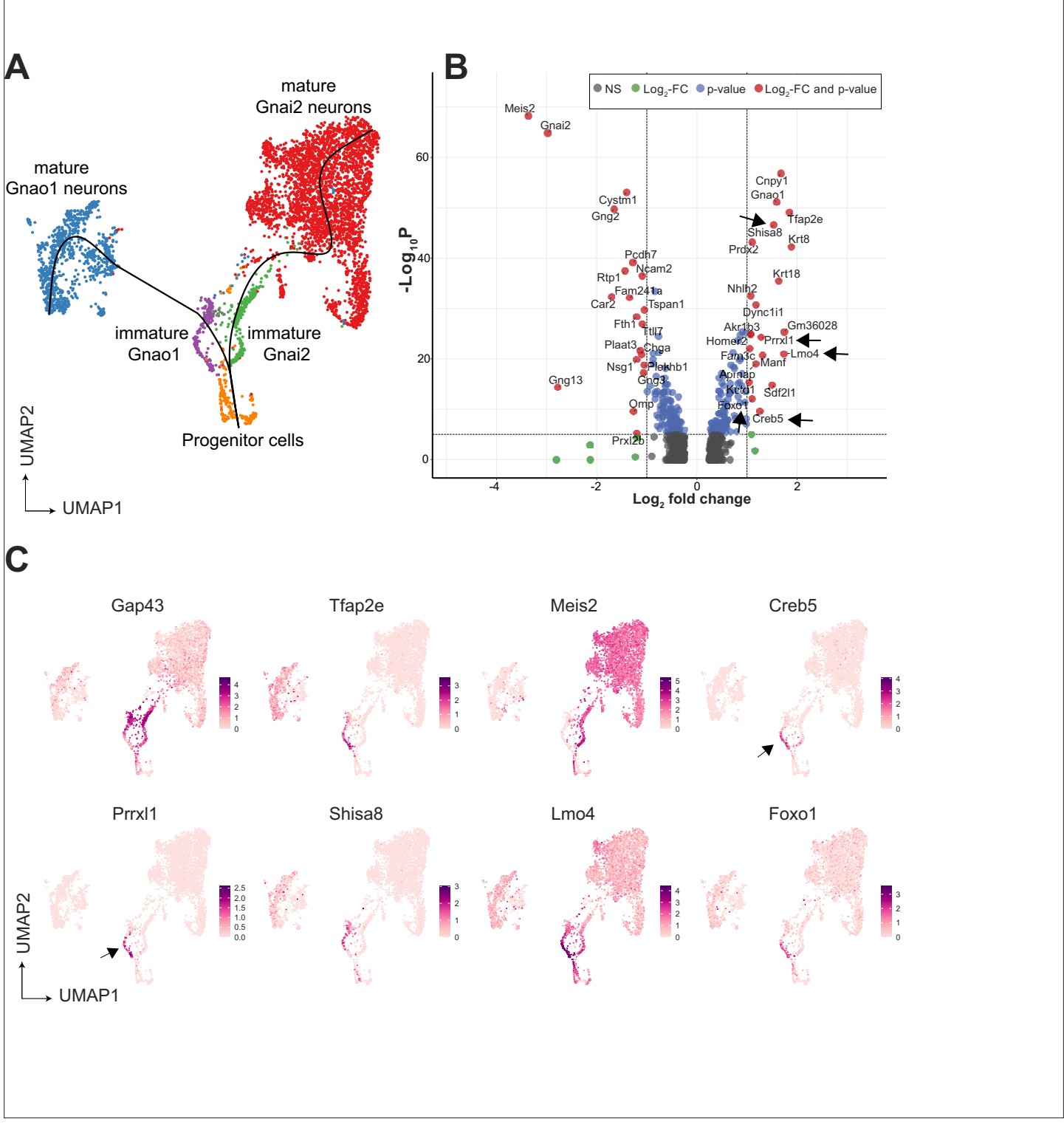

**Figure 3.** Gene expression dynamics during development of sensory neurons. (**A**) Uniform Manifold Approximation and Projection (UMAP) of neuronal cells with clusters (n1-n13 from *Figure 3—figure supplement 1*) represented in different colors. Sub-clusters within mature Gnai2, and Gnao1 neurons are merged. The line on the UMAP plot shows the pseudotime developmental trajectory of neurons. (**B**) Volcano pot of differential gene expression between immature Gnai2 (cluster n6; Gap43+, Gnai2+) vs immature Gnao1 (cluster n7; Gap43+, Gnao1+) neurons. Genes that satisfy |log2 fold change normalized expression|>1 (green) and -log10 p-value >6 (blue) are considered significant and colored in red. Non-significant (NS) genes are colored in gray. Arrows point to transcription regulators enriched in Gnao1 + immature neurons. Complete list of differentially expressed genes is in *Figure 3—source data 1*. (**C**) Feature plots showing the normalized expression levels of previously known markers for immature neurons (Gap43), Gnao1 neurons

*Figure 3 continued on next page*

*Figure 3 continued*

(Tfap2e), Gnai2 neurons (Meis2), and transcription factor or related genes: Creb5, Prrxl1, Shisa8, Lmo4, Foxo1. Arrows highlight the limited expression of Creb5 and Prrxl1 to immature neurons, but are absent from mature indicating transient expression during the development of Gnao1 neurons. Subclusters within Gnao1, Gnai2 neurons and the effect of VRs on sub-clustering are in *Figure 3—figure supplements 1 and 2* respectively.

The online version of this article includes the following source data and figure supplement(s) for figure 3:

**Source data 1.** Differentially expressed genes between immature Gnao1 and immature Gnai2 neurons.

**Figure supplement 1.** Neuronal cell types of vomeronasal sensory epithelium.

**Figure supplement 2.** Effect of VR genes on neuronal clustering.

other family-C members (*Vmn2r3*, *Vmn2r4*, *Vmn2r5*, *Vmn2r6*, *Vmn2r7*) (*Figure 4C and E*). Cluster n2 is a mixture of either *Vmn2r1* or *Vmn2r2* expressing neurons (*Figure 4E*), with *Vmn2r2* being similarly co-expressed along with other family-C members in those cells. We observed that among family-ABDE members, A1-A6 V2Rs are highly expressed in cluster n2, n3 which co-express Vmn2r2 whereas A8-A10, B, D, and E family V2Rs are mainly expressed along with *Vmn2r1* in cluster n1 neurons (*Figure 4E*). These observations of family biased expression of ABDE with *Vmn2r1* or *Vmn2r2* agrees with earlier studies (*Silvotti et al., 2007*; *Silvotti et al., 2011*) that have used antibody or RNA probe based approaches.

Furthermore, we observed that expression of H2-Mv genes (*H2-M10.1, H2-M10.2, H2-M10.3, H2-M10.4, H2-M10.5, H2-M10.6*) is largely confined to clusters n2 and n3, where they are co-expressed selectively in *Vmn2r2* positive neurons and mostly absent from *Vmn2r1* expressing cluster n1, n2 neurons (*Figure 4D and E*, *Figure 4—figure supplement 2A*). These observations align with prior reports from antibody or RNA probe-based experiments, that demonstrated a sub-division amongst Gnao1 neurons based on their expression of family-C Vmn2rs and co-expression of H2-Mv genes with the *Vmn2r2* subset (*Silvotti et al., 2007*).

We then asked whether, at the single neuron level, we could identify trends in co-expression patterns between ABD-family V2Rs and H2-Mv genes, or between members of the ABD family themselves. We performed co-expression analysis to identify specific VR and H2-Mv combinations by setting an expression threshold from normalized counts. The distribution of normalized expression for VRs and H2-Mvs across cells was bimodal, with the first peak near zero and a second peak likely corresponding to true expression (*Figure 4—figure supplement 1D–F*). We set the starting of the second peak as a threshold to call the expression of VRs or H2-Mvs in a single-cell. Thus, a VR or H2-Mv was considered as co-expressed in a cell only if its normalized expression value surpasses the set threshold based on the distribution (*Figure 4—figure supplement 1E and F*). The threshold was 1.25 for V2Rs (*Figure 4—figure supplement 1E*) and H2-Mv genes (*Figure 4—figure supplement 1F*). This analysis resulted in the identification of multiple co-expressing V2Rs and H2-Mv genes per cell. The chance that a combinatorial pattern identified can be due to a doublet is dependent on the frequency of expression of that gene. Therefore, for abundantly expressed family C V2Rs, it is possible some combinatorial patterns may be from doublets. But in the case of family ABD V2Rs, the probability of a particular combination falsely identified due to a doublet more than once is very low. Hence, we assigned a threshold for combinatorial co-expression patterns that are identified in two or more cells to be more reliable. These combinatorial expression patterns along with their observed frequency are listed in *Supplementary file 5*.

Within the expression patterns represented by a frequency of more than two cells, we observed that family C1 V2R (*Vmn2r1*) is mostly expressed alone without any other family-C members (*Figure 4F and E*) as shown earlier (*Silvotti et al., 2007*). Surprisingly a few cells (31 neurons out of 1025 Gnao1 neurons) were observed to co-express *Vmn2r1* along with *Vmn2r7* (*Supplementary file 5*), contrary to earlier reports that Vmn2r1 does not co-express with family-C2 receptors. Most of Gnao1 neurons with family C2 V2Rs (*Vmn2r2-Vmn2r7*) co-express multiple members amongst them, ranging from 2 to 6 V2Rs per cell (*Figure 4F and E*).

In the case of family-ABD V2R expression, we observed that most Gnao1 neurons follow the 'one ABD-V2R, one neuron' rule, except for a few that express two ABD-V2Rs (*Figure 4G*). Amongst cells that express more than two ABD-V2Rs, the combination of *Vmn2r20* and *Vmn2r22* stood out as the highest (44 cells), followed by *Vmn2r39*, *Vmn2r43*, and *Vmn2r50* (10 cells, *Supplementary file 5*). It is worth noting that these ABD co-expressions in each cell are observed irrespective of whether that cell

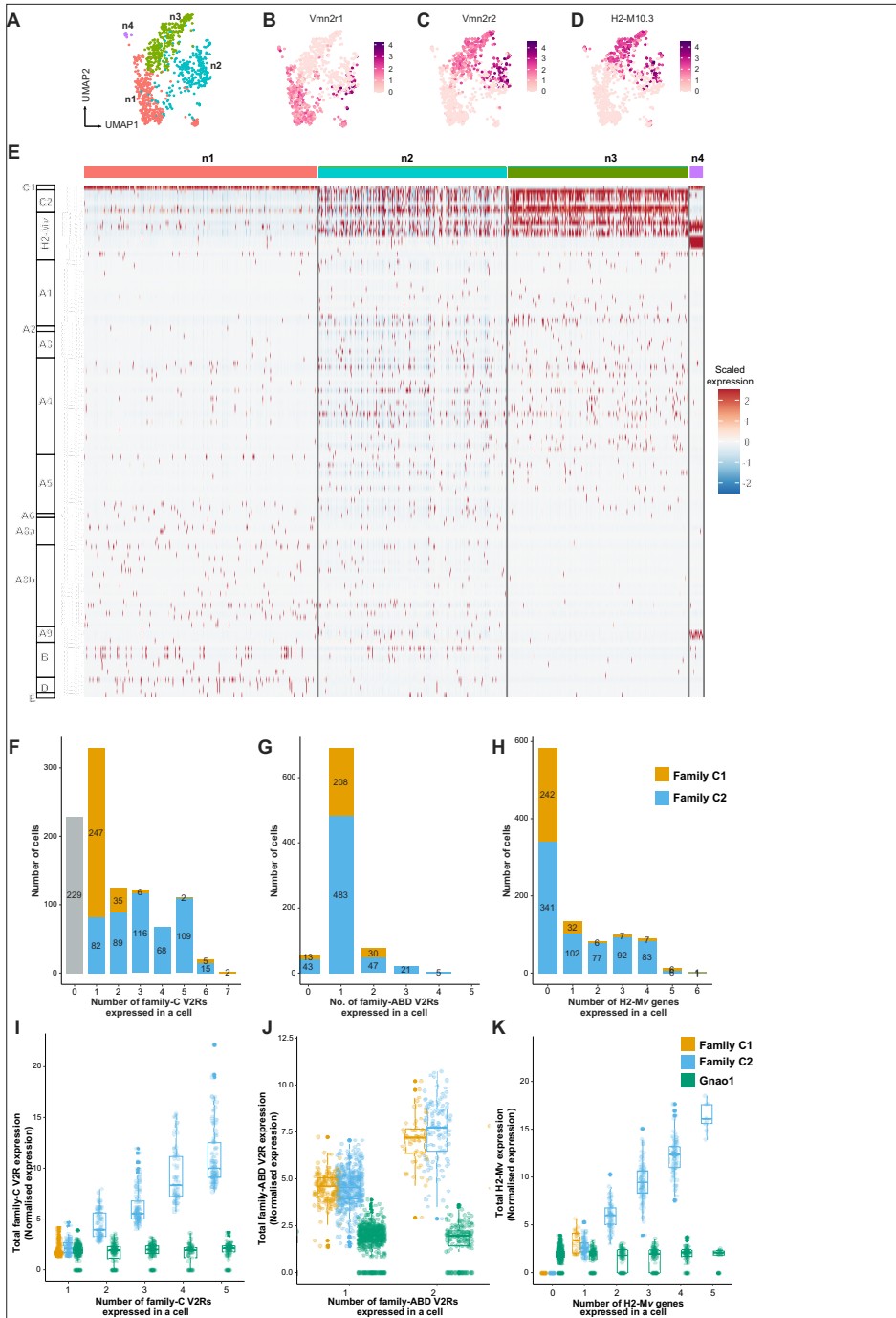

**Figure 4.** Subpopulation of Gnao1 neurons defined by vomeronasal type-2 GPCR (V2R) and H2-Mv expression. (**A**) Uniform Manifold Approximation and Projection (UMAP) representation of Gnao1 neurons from *Figure 3*. Each dot represents a cell and four Gnao1 neuron clusters (**n1–n4**) are marked in different colors. (**B–D**) Feature plot showing exclusive expression of Vmn2r1 (**B**), Vmn2r2 (**C**), and the most abundant H2-Mv gene, H2-M10.3 (**D**) in Gnao1 neurons. (**E**) Heat map showing the expression of V2R and H2-Mv genes in Gnao1 neurons. Cluster numbers are marked on the top with color coding as in (**A**) and gene families are labeled on the left. Each column represents a cell and the scaled gene expression in each row is color coded as per the scale with red and blue indicating high and low expression, respectively. Vmn2r1 is expressed in almost all cells of cluster-1 and cluster-4; Vmn2r2 is expressed in all cells of Cluster 3; Cluster2 has mutually exclusive expression of Vmn2r1 and Vmn2r2. (**F–H**) Bar plot showing a number of cells expressing: 0–7 family-C vomeronasal type-2 GPCRs (V2Rs) per cell (**F**), 0–5 family-ABD V2Rs per cell (**G**), 0–6 H2-Mv genes per cell (**H**) with composition of cells associated with family C1 (orange) or C2 (blue) V2R color coded on the bar. (**I**) Box plots comparing the sum of normalized expression levels

*Figure 4 continued on next page*

*Figure 4 continued*

of family-C V2Rs and Gnao1 (Green) in a cell that expresses 1–5 family-C V2Rs. (**J**) Box plot comparing the level of total ABD-V2R expression from cells with 1 and 2 ABD-VRs along with Gnao1 expression level (green). (**K**) Box plot comparing the level of total H2-Mv expression in cells that express 1–5 H2-Mv genes along with Gnao1 expression level (Green). Multiple combinations of family-C, family ABD V2Rs, and H2-Mvs identified to be co-expressed in a single-cell and their cell frequency are listed in **Supplementary file 5**.

The online version of this article includes the following figure supplement(s) for figure 4:

**Figure supplement 1.** Frequency and distribution of vomeronasal type-2 GPCR (V2R) and H2-Mv genes.

**Figure supplement 2.** Characteristics of H2-Mv expression.

**Figure supplement 3.** Co-expression characteristics of vomeronasal type-2 GPCR (V2R) and H2-Mv genes using data from *Hills et al., 2024*.

**Figure supplement 4.** Co-expression of vomeronasal type-I GPCRs (V1Rs) in Gnai2 neurons.

---

expresses family-C1 or C2 V2R (*Figure 4G*). Among Gnao1 neurons (n1-n4), 229 cells did not express family-C V2Rs (*Figure 4F*) and 46 cells did not express ABD-V2Rs (*Figure 4G*) as per our threshold.

In the case of H2-Mv genes, each cell expresses 1–4 members mostly with famlily-C2 V2Rs (*Figure 4H, C and D*). Interestingly, in deviation from this, we observed a pattern whereby *H2-M1*, *H2-M9,* and *H2-M11* genes that show sequence divergence from rest of the H2-Mv cluster (*Ishii et al., 2003*), are specifically expressed in cluster n4 and restricted cells amongst cluster n3, where they co-express with Vmn2r1 GPCR, rather than family-C2 V2Rs (*Figure 4—figure supplement 2B and D*). Overall, 56.9% of Gnao1 cells that express either C1-V2R or C2-V2R, did not express any H2-Mv genes (*Figure 4H*) as per our threshold, reinforcing earlier observations (*Ishii and Mombaerts, 2008*) that their functional role maybe restricted to a subset of V2R expressing neurons. Furthermore, these *H2-M1*, *H2-M9,* or *H2-M11* expressing cells along with Vmn2r1, also co-express either *Vmn2r81* or *Vmn2r82* (V2rf) (*Figure 4—figure supplement 2B and C*) as identified before (*Ishii et al., 2003*). Two color RNA in situ hybridization with a common probe targeting *Vmn2r81* and Vmn2r*82* and separate probes for *H2-M1*, *H2-M9,* and *H2-M11* shows that Vmn2r81/82 cells always colocalized with the selected H2-Mv probes (*Figure 4—figure supplement 2E*).

We performed a similar co-expression analysis on scRNAseq data of p56 VNO from *Hills et al., 2024* and observed that the overall co-expression pattern of V2Rs and H2-Mv genes in Gnao1 neurons matches with ours (*Figure 4—figure supplement 3*). This rules out the possibility of dataset-specific artifacts leading to spurious co-expression patterns. Altogether, these data validate the co-expression analysis methodology and identify novel combinatorial co-expression patterns of V2Rs and H2Mv genes.

We next asked, whether the total GPCR or H2-Mv expression level is regulated or capped in a single-cell? To answer this question, we calculated the total family-C-V2R, ABD-V2R, and H2-Mv expression in a single-cell that either expresses one member or multiple members of each family (*Figure 4I–K*). The results showed that when cells express more than one V2R, total V2R expression of either family-C or family-ABD (*Figure 4I and J*) scales up proportional to a total number of expressed V2Rs of that family. This is also true in the case of multiple H2-Mv expression where total H2-Mv expression is proportional to the number of H2-Mv genes expressed in a cell (*Figure 4K*). As a control, and to eliminate the possibility of doublets, we see that the levels of Gnao1 do not change with a number of V2Rs or H2-Mv genes expressed in a cell (*Figure 4I–K*).

Lastly, we performed a similar co-expression analysis for V1Rs with a threshold value of 2.5 (*Figure 4—figure supplement 1D*). In similarity to the pattern observed for ABD-V2Rs, most Gnai2 neurons expressed a single V1R per cell with some cells co-expressing two V1Rs (*Figure 4—figure supplement 4B and A*, *Supplementary file 6*). Some of these V1R combinations have been reported recently (*Lee et al., 2019*). Even in the case of Gnai2 neurons, total V1R expression in the cells that co-express two receptors is higher compared to cells that express one GPCR with Gnai2 levels being same (*Figure 4—figure supplement 4C*), indicating that these measurements are from singlet cells. Due to the stringent expression threshold, it is possible that VRs expressed at very low levels were not considered and thereby leading to neurons where zero VRs are detected (*Figure 4—figure supplement 4B*). However, these neurons express other genes like neuronal marker Gnai2 at same level as other neurons (*Figure 4—figure supplement 4C*) ruling out the possibility of them being debris.

Collectively, these data reveal the patterns of GPCR, and H2-Mv co-expression in VNO neurons, which is likely to be instrumental in deciphering how these neurons respond to a single or combination of stimuli, as well as the cellular mechanisms that orchestrate these expression patterns.

## Divergent gene expression profile of mature Gnao1 and Gnai2 neurons

Since the exclusion of VR genes did not fundamentally alter the clustering and divergence into mature Gnai2 /Gnao1 neurons, we decided to investigate the gene expression differences between these neurons by performing differential gene expression analysis of mature Gnao1 (cluster 14 from *Figure 1*) and Gnai2 neurons (cluster 7, 16 from *Figure 1*) and identified 924 differentially expressed genes with at least log2 fold change of 1 and adjusted p-value <10e-6. Of these, 456 genes are enriched in Gnao1 neurons and 468 genes in Gnai2 neurons (*Figure 5A*). In agreement with previous studies, we see enrichment of V2Rs, H2-Mvs, and *B2m* (*Ishii et al., 2003*; *Loconto et al., 2003*), *Robo2* (*Prince et al., 2009*), *Tfap2e* (*Lin et al., 2018*), *Calr4* (*Dey and Matsunami, 2011*) in Gnao1 neurons and V1Rs, *Meis2* (*Chang and Parrilla, 2016*) in Gnai2 neurons (*Figure 5A*). Along with such known genes, our analysis revealed new differentially expressed genes in the two neuronal subtypes.

To confirm differential gene expression amongst mature neurons, we performed two-color fluorescence RNA-ISH of the top candidates from *Figure 5A*, using *Gnao1* or *Gnai2* as co-labeling markers (*Figure 5B, C and D*). Among these genes, *Nrp2* and *Socs2* expression is restricted to Gnai2 neurons (*Figure 5C*); *Tppp3* and *Gm36028* expression is restricted to Gnao1 neurons (*Figure 5D*). Some of the enriched genes such as *Ckb* in Gnai2 and *Krt18* in Gnao1 neurons, were also observed to be expressed in sustentacular cells (*Figure 5C and D*). Furthermore, from chromogenic RNA-ISH experiments, we observed genes such as *Nsg1*, *Rtp1*, *Dner*, *Qpct*, *Pcdh7* to be either restricted to or highly expressed in Gnai2 neurons (*Figure 5—figure supplement 1A*) and others such as *Apmap*, *Selenof*, *Hspa5*, *Itm2b*, *Agpat5* were observed to have a gradient of high to low-level expression from *Gnao1* to *Gnai2* zones (*Figure 5—figure supplement 1B*). Two genes: *Sncg* and *Prph* specifically expressed in scattered subsets of Gnao1 or Gnai2 neurons, respectively, (*Figure 5—figure supplement 1A and B*) indicating selective expression in a particular neuronal subpopulation. Gene ontology terms associated with the genes validated with RNA-ISH are in *Supplementary file 7*.

These experiments validated that several of the candidate genes identified as differentially expressed from scRNA seq data, are highly specific and spatially localized to one of the neuronal subpopulations. Taken together our data indicate that VNO neurons that start from a common progenitor, go on to transiently express genes during immature stages that could decide their cell fate and these cells exhibit markedly different gene expression profiles in the mature stage.

## Differential ER environment in Gnao1 and Gnai2 neurons

Given that Gnao1 and Gnai2 neurons express divergent GPCR families, we asked the question of whether the differentially expressed genes indicate fundamental differences in cellular or molecular processes between these two cell types. We performed gene ontology (GO) enrichment analysis on the two sets of genes. Interestingly, we observed that amongst the Gnao1 neuron genes, most of the enriched terms with p-value <0.05, are related to endoplasmic reticulum (ER) processes, which include 'protein folding,' 'response to ER stress,' and 'ERAD pathway' (*Figure 6A*). Conversely, a gene ontology analysis of Gnai2 neuron genes did not reveal any particular enrichment of GO terms in this neuronal subset. As more than 20% of the genes associated with protein folding (*Figure 6A*) are enriched in Gnao1 neurons, we sought to probe the spatial transcription pattern of some of the candidates. Fluorescence RNA in-situ hybridization confirmed the selective expression of ER genes such as *Creld2*, *Dnajc3*, *Pdia6*, and *Sdf2l1* amongst Gnao1 neurons (*Figure 6B*).

To test whether these findings extend to the protein level, we performed immuno-fluorescence microscopy of VNO sections using antibodies specific to a collection of ER proteins. We first tried an antibody against SEKDEL, a common ER retention signal, which intensely labeled Gnao1 neurons (*Figure 7A*). This selective labeling was confirmed by co-labeling with anti-Gnao1 (*Figure 7B and D*), where an anti-SEKDEL ER signal is seen in Gnao1 neurons. We quantified enrichment of SEKDEL across multiple (26) VNO sections from three mice, labeled with anti-SEKDEL, anti-Gnao1, and anti-Omp as a control. Normalized fluorescence intensity quantified along the apical-basal axis (*Figure 7D and E*), shows that SEKDEL signal intensity increases along with Gnao1. Surprisingly, several other antibodies that detect ER chaperone proteins: Hspa5 (BiP), PDI, Grp94, Calnexin (*Figure 7F*, *Figure 7—figure*

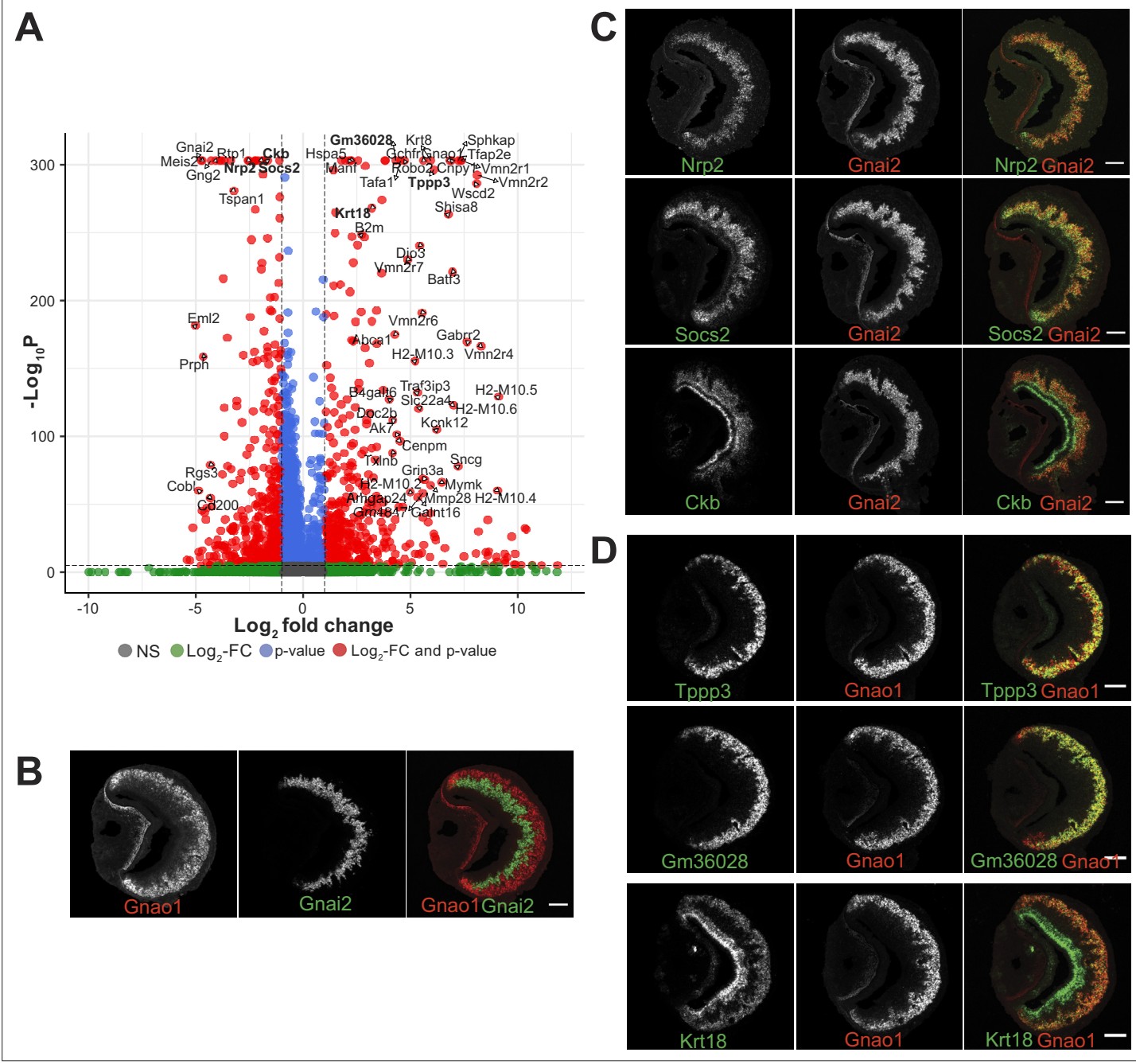

**Figure 5.** Divergent gene expression: Gnao1 vs Gnai2 neurons. (**A**) Volcano plot showing differentially expressed genes between mature Gnao1 and Gnai2 neurons. Genes that satisfy |log2 fold change normalized expression|>2 (green) and -log10 p-value >6 (blue) are considered significant and colored in red. Non-significant (NS) genes are colored in gray. Complete list of differentially expressed genes is in *Figure 5—source data 1*. (**C–D**) Two color RNA in situ hybridization (ISH) of selected enriched genes marked in bold on the volcano plot. Gnao1, Gnai2, respective markers of basal and apical neurons are shown in (**B**). Genes enriched in Gnai2 neurons (**C**) or Gnao1 neurons (**D**) are co-localized with the respective markers. RNA-ISH for additional enriched genes is shown in *Figure 5—figure supplement 1*. Scale bar:100 μm.

The online version of this article includes the following source data and figure supplement(s) for figure 5:

**Source data 1.** Differentially expressed genes between mature Gnao1 and mature Gnai2 neurons.

**Figure supplement 1.** Expression pattern of enriched genes in Gnai2 or Gnao1 neurons.

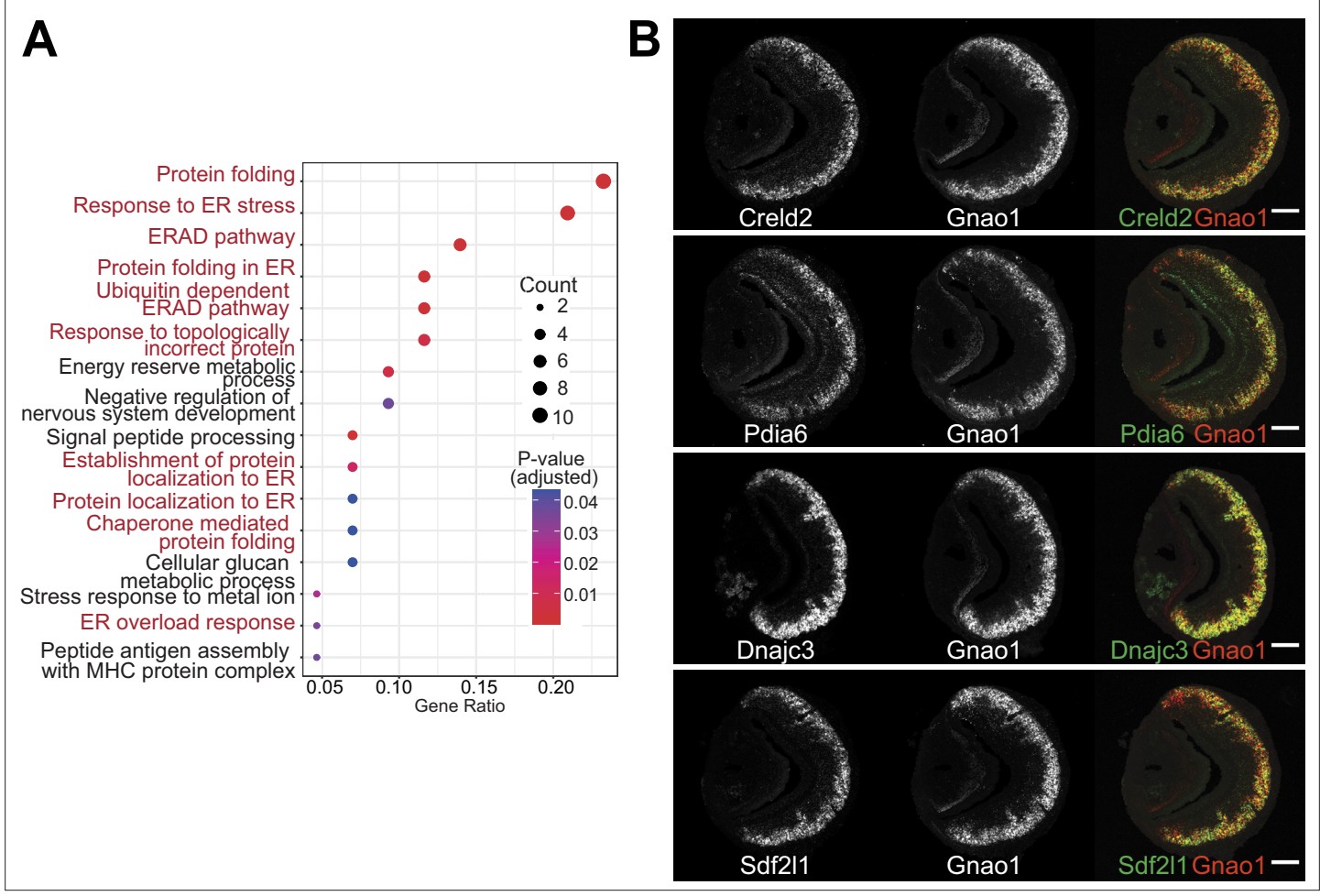

**Figure 6.** Enrichment of endoplasmic reticulum (ER) genes in Gnao1 neurons. (**A**) ER gene expression in Gnao1 neurons. Annotation of gene ontology (GO) biological processes of genes that are significantly enriched in Gnao1 neurons from *Figure 5A*. GO terms related to ER processes are marked in red. p-value <0.05 was used as a cut-off. (**B**) Vomeronasal organ (VNO) coronal section two-color fluorescent RNA-in situ hybridization (ISH) of selected Gnao1 enriched ER chaperone genes (Creld2, Pdia6, Dnajc3, Sdf2l1: green), co-labeled with Gnao1 probe (red) shows Gnao1 zone restricted expression of these genes. Scale bar 100 µm.

supplement 1A); as well as antibodies against ER membrane/structural proteins: Sec61β, Atlastin1, NogoB, Climp63, Reep5 (*Goyal and Blackstone, 2013*) show a similar enrichment in Gnao1 neurons, as seen from the immune-fluorescence images and their quantification with Gnao1 or SEKDEL antibodies (*Figure 7G and H*, *Figure 7—figure supplement 1B*). Some ER protein localizations (SEKDEL, Hspa5, PDI, Atlastin1, NogoB) were detected selectively amongst basal zone Gnao1 neurons, while others (Sec61β, Calnexin, Grp94, Reep5, Climp63) detected a pattern of Gnao1 neuron enrichment, with lower levels in Gnai2 neurons. To test whether higher levels of these ER proteins resulted from their increased transcription, we compared their RNA levels in Gnai2 /Gnao1 neurons from our scRNA dataset (*Figure 7—figure supplement 2*). In the case of Hspa5 (Bip), Hsp90b1 (Grp94), and Sec61b mRNA levels are higher in Gnao1 neurons (also see Hspa5 in *Figure 5A*, and RNA-ISH in *Figure 5—figure supplement 1*). On the other hand, RNA levels of Atlastin1, and PDI were comparable (*Figure 7—figure supplement 2*), indicating that Gnao1 neuron enrichment of some ER proteins could also arise due to post-transcriptional regulatory mechanisms.

Many of the Gnao1 enriched ER genes and their protein products are chaperones, while others are ER structural proteins. The upregulation of both types of ER proteins prompted us to examine the cells by electron microscopy. Serial sections (70 nm thick) of ~1 mm square regions of the VNO were collected on tape and were imaged by scanning electron microscopy (*Baena et al., 2019*). Since the sections contained the entire VNO, we were able to distinguish Gnai2 from Gnao1 cells by their

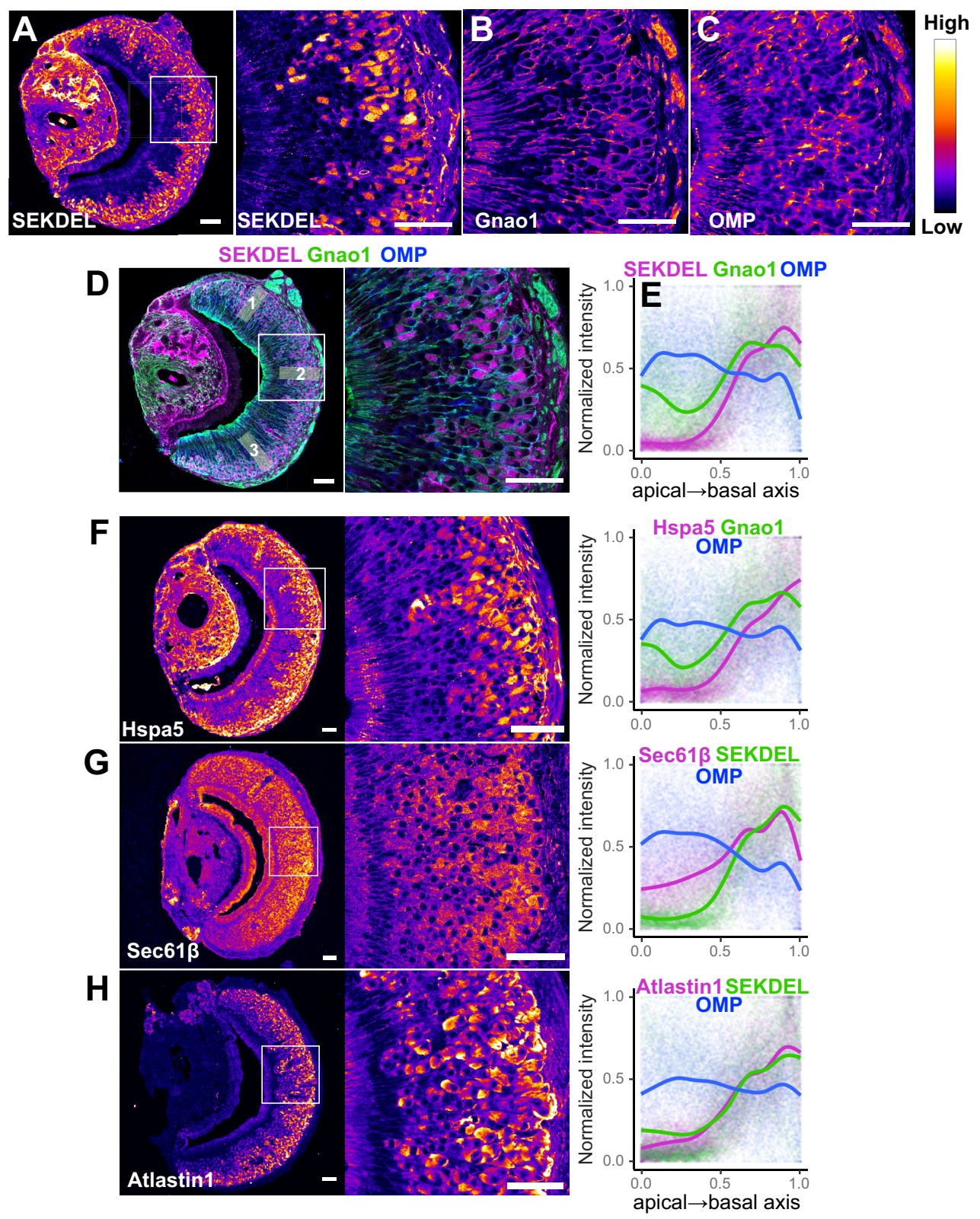

**Figure 7.** Differential localization of endoplasmic reticulum (ER) proteins in Vomeronasal organ (VNO) neurons. Pseudocolored immunofluorescence images of VNO coronal sections labeled with antibodies against KDEL (anti-SEKDEL), a common ER retention signal (**A**). The section is co-labeled with anti-Gnao1 to mark basal zone neurons (**B**) and anti-OMP to mark all mature neurons (**C**). Signal intensity of KDEL, Gnao1, and OMP channels are quantified from ROIs along the apical-basal axis as shown in example (**D**). Signal intensity measured from multiple sections (n>20 for each antibody)

*Figure 7 continued on next page*

*Figure 7 continued*

from three biological replicates was normalized and the trendline was fitted to show the Gnao1 neuron-enriched localization of anti-KDEL (**E**). Points on the plot show normalized intensity values color-coded for each antibody on which the trendline was fitted. Similar immuno-labeling and quantification of ER chaperone Hspa5 (BiP) (**F**), ER membrane translocon subunit Sec61β (**G**), and ER membrane protein Atlastin1 (**H**) indicate their enrichment in Gnao1 neurons compared to Gnai2 neurons. Distribution of additional ER chaperone and membrane proteins is shown in *Figure 7—figure supplement 1* along with combined source data in *Figure 7—source data 1*. Scale bar: 50 µm.

The online version of this article includes the following source data and figure supplement(s) for figure 7:

**Source data 1.** Fluorescence intensity values of ROIs along an apical-basal axis from immunolabeled images of Vomeronasal organ (VNO) neuroepithelium.

**Figure supplement 1.** Pseudocolored immunofluorescence images of Vomeronasal organ (VNO) coronal sections labeled with antibodies against endoplasmic reticulum (ER) proteins.

**Figure supplement 2.** Comparison of endoplasmic reticulum (ER) gene expression between Gnai2, Gnao1 neurons.

location as they are spatially segregated into apical or basal zones, respectively, within the neuroepithelium (*Figure 8A*). In Gnao1 cells, more than half of the cytoplasm contained dense smooth ER with cubic membrane morphology (*Almsherqi et al., 2006*; *Almsherqi et al., 2009*), also known as organized smooth ER (OSER) (*Snapp et al., 2003*; *Figure 8A and B*, *Figure 8—figure supplement 1*). In the remainder of the cytoplasm, there were closely apposed parallel sheets of ER membrane that were contiguous with lamellar ER surrounding the nucleus (*Figure 8—figure supplement 1*). The apical/Gnai2 neuronal cell bodies are smaller than those of Gnao1 cell bodies. They also appeared to have cubic ER membranes, but these seemed denser and occupied a smaller fraction of cytoplasmic volume (*Figure 8A and C*).

Taken together, our data indicates that mature Gnao1 neurons differ substantially from Gnai2 neurons in selectively upregulating the transcription of several ER genes. Furthermore, this neuronal subset also exhibits higher levels of ER proteins, via transcriptional and post-transcriptional mechanisms, as well as a hypertrophic smooth ER that is arranged in the form of an organized cubic membrane ultrastructure. Collectively, these observations indicate a specialized ER environment that could play a significant functional role in the Gnao1 subset of VNO neurons.

## Discussion

The vomeronasal organ has been a model of considerable interest for molecular sensory biology, to study the diversification of sensory neurons, uniquely evolved gene families, and their patterns of expression, all of which are essential to understand how specific social chemo-signals elicit innate behaviors. In this study, using single-cell and low-level spatial transcriptomics, we developed a comprehensive resource identifying cell types of the VNO neuroepithelium and studied the developmental and receptor co-expression patterns within sensory neurons. We organized our dataset as an interactive web portal resource, accessible from https://www.scvnexplorer.com, where a gene can be queried for its expression pattern and levels amongst cell clusters.

The identification of distinct cell clusters and validation of marker expression through RNA in situ hybridization, revealed a heterogeneous cellular composition of the vomeronasal neuroepithelium. In particular, we identified specific new gene markers, such as *Ppic* in sustentacular cells and *Ly6d* in non-sensory epithelium, which would be helpful in the future to genetically tag or target these cell types and to study their developmental origins. Our study also identified specific types of macrophage cells within the neuroepithelium. Given the increasing appreciation of the role of tissue-resident or infiltrating macrophages in a variety of physiological processes (*Nobs and Kopf, 2021*; *Mass et al., 2023*) it is tempting to speculate that macrophages could also play an important role in VNO neuronal differentiation or the regulation of inflammatory/immune responses to external insults. Future experiments, performing targeted purification, deep sequencing, and functional manipulation of VNO macrophage types could provide deeper insights into their role in VNO function and whether they share cellular identity with MOE macrophages.

### Development of VSNs

VNO sensory neurons expressing *Gnao1* and *Gnai2* develop continuously throughout life from a single progenitor cell population located within the VNO marginal zones and differentiate into immature and

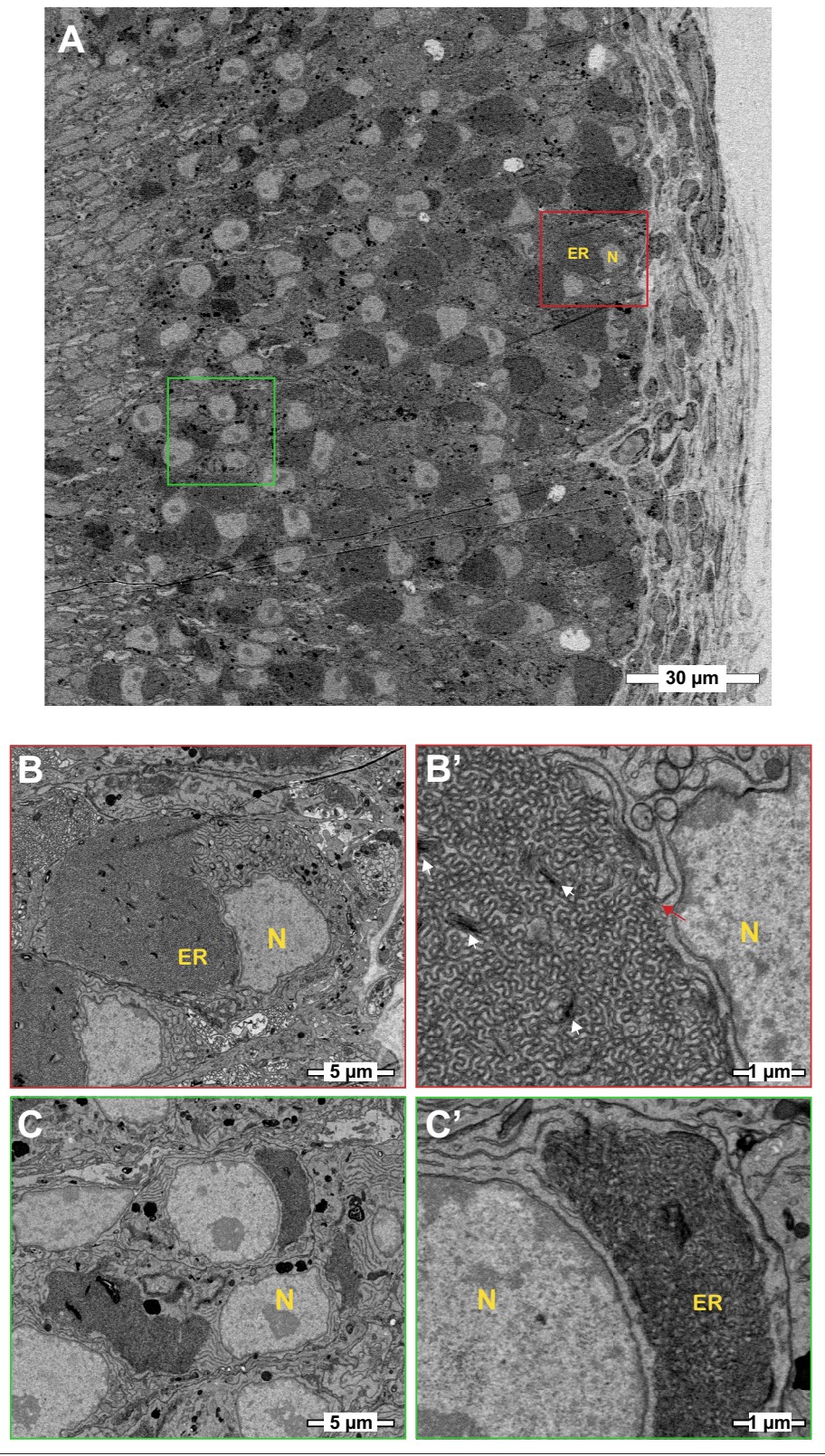

**Figure 8.** Basal/Gnao1 neurons are densely packed with cubic membrane endoplasmic reticulum (ER). (**A**) Scanning electron micrographs of the vomeronasal sensory epithelium at low magnification showing cell bodies of VNO sensory neurons (VSNs), sustentacular cells, and basal lamina. Boxed regions in red or green mark cell bodies of basal/Gnao1 or apical/Gnai2 neurons, respectively, that are displayed at higher magnification below.

*Figure 8 continued on next page*

*Figure 8 continued*

Nucleus (**N**) appears light and ER is dark. Cell bodies of basal/Gnao1 neurons are larger and occupied by a substantial amount of ER, in comparison to apical/Gnai2 neurons. (**B, B'**) Magnified micrographs show the cell body of a basal/Gnao1 neuron, packed with cubic ER membranes. White arrows point to dense membrane stacks that are better resolved in *Figure 8—figure supplement 1*. Red arrow points to the lamellar ER membrane that is contiguous with the cubic membrane. (**C, C'**) Cell bodies of apical/Gnai2 neurons also seem to show dense cubic membrane ER, that is smaller in comparison to basal/Gnao1 neurons.

The online version of this article includes the following figure supplement(s) for figure 8:

**Figure supplement 1.** Cubic membrane endoplasmic reticulum (ER) in Gnao1 neurons.

mature neurons. Recent developmental studies have demonstrated that the dichotomy is established by Notch signaling followed by transcription factor, *Bcl11b* expression that marks the progenitor cells to take Gnao1 fate (*Enomoto et al., 2011*; *Katreddi et al., 2022*). Most Importantly, a critical period of Notch signaling determines whether progenitors give rise to apical, basal, or sustentacular cell types. After Gnao1/Gnai2 identity is established, transcription factor *Tfap2e*, continuously expressed from immature to mature Gnao1 neurons, is required to maintain the Gnao1 fate (*Lin et al., 2018*; *Lin et al., 2022*; *Katreddi et al., 2022*). Pseudo-time analysis of our data confirmed a developmental trajectory that starts with progenitor cells, leading to immature neurons that take on either Gnai2 or Gnao1 fate. From our scRNA seq data, a comparison of immature neurons revealed that apart from *Tfap2e*, transcription factors: *Creb5*, *Prrxl1*, *Lmo4*, and *Foxo1* also express in immature Gnao1 neurons. But unlike *Tfap2e*, *Creb5*, and *Prrxl1* expression is restricted to only the immature stage of Gnao1 neurons. Although, *Foxo*1, and *Lmo4* continue to express in mature Gnao1 neurons, their RNA levels are high in immature Gnao1 neurons and reduce upon maturation. Among the transcription factors identified, *Prrxl1* has been shown to be involved in the development of dorsal root ganglion neurons where it has been shown to autoregulate its expression (*Monteiro et al., 2014*). Our observations on the tight temporal regulation of these transcription factor-related genes in the developmental trajectory of Gnao1 neurons indicate that additional molecular players might be important in further specification or maintenance of this neuronal lineage. Further experiments are needed to tease out the mechanism and precise choreography by which these transcription factors collectively specify and maintain the Gnao1 cell identity as well as gene expression patterns unique to these neurons.

## Receptor co-expression in Gnao1 neurons

One of the key features distinguishing basal zone Gnao1 VNO neurons from apical Gnai2 or MOE neurons, is the combinatorial co-expression of two or more V2R family GPCRs along with H2-Mv class-1b MHC molecules in each cell. Our analysis of receptor expression in mature VNO neurons, revealed that members of V1R and V2R-ABD gene families largely follow the one-neuron-one receptor rule, but with multiple exceptions. Some V1Rs were found to consistently co-express in Gnai2 neurons and the major combinations observed from our analysis (such as Vmn1r85/86, *Figure 4—figure supplement 4*), match those identified recently from a functional single-cell study (*Lee et al., 2019*), leading confidence to the methodology used in our co-expression analysis. Receptors of the ABD-V2R subfamilies are also mostly expressed at one receptor per neuron, with notable exceptions being the observed combinations of *Vmn2r20/22* and *Vmn2r39/43/50* being co-expressed per cell.

In our study, the analysis of family-C, family ABD-V2Rs and H2-Mv genes expressed per cell, confirmed the earlier reported sub-division of basal/Gnao1 neurons into those that express either *Vmn2r1* (family-C1) or *Vmn2r2-Vmn2r7* (family-C2), resulting in a broad four-part division of mature VNO neurons: those expressing (a) *Gnai2*/V1Rs, (b) *Gnao1*/*Vmn2r1*, (c) *Gnao1*/*Vmn2r2-2r7*; H2-Mv+, (d) *Gnao1*/*Vmn2r2-2r7*; H2-Mv-. The co-expression of ABD-V2R sub-families A1-A5 and most of the H2-Mv genes with *Vmn2r2*/family-C2 neurons (*Figure 4C–E*, **cluster n3, n2,** *Supplementary file 5*) as well as preferential co-expression of A6-A9, B, D, E sub-families with Vmn2r1 along with exclusion of H2Mv genes from these neurons (*Figure 4E*), points to the overall non-random logic of V2R/H2Mv co-expression. It would be interesting to see how receptor deorphanization and future functional experiments will map onto these elaborate co-expression patterns. At the same time, there were notable deviations observed from these rules, namely the restricted and sparse expression of *H2-M1/*

*M9/M11* with *Vmn2r1* neurons in cluster n4 (*Figure 4E*, *Figure 4—figure supplement 2B and E*), where the receptors *Vmn2r81/82* are also expressed selectively in these neurons.

Of note, the combined expression of VR/H2-M*v* in a cell is not capped and is proportional to a number of VRs and H2-Mv genes expressed in that cell (*Figure 4I–K*). It would be interesting to see whether the levels of co-expressing transcripts also translate to protein levels in cells and their impact on functional interactions, if any, between co-expressed proteins.

## ER environment in VNO neurons

When comparing gene expression amongst Gnai2/Gnao1 mature neurons, what stood out was the unexpected enrichment of gene ontology terms associated with ER functions within Gnao1 neurons (*Figure 6A and E*). Most of these ER-Gnao1 enriched genes from our dataset, match the ones identified from a similar comparison by *Lin et al., 2022*, supporting our findings. We validated several of these ER genes via RNA-ISH to be exclusively expressed or enriched in Gnao1 neurons. Even more puzzling was our observation that this enrichment pattern extended at the protein level to many ER proteins probed by immune labeling (*Figure 7*). In some cases, both RNA and proteins were Gnao1 enriched, whereas in others, RNA levels were comparable but protein levels were biased towards Gnao1 cells. The enrichment of ER genes/proteins via transcriptional and/or post-transcriptional mechanisms presents a new feature of these cell types that could be associated with their differentiation and function. High levels of several chaperones such as Creld2, Dnajc3, Pdia6, PDI, Hspa5/Bip, Grp94, and Calnexin in Gnao1 neurons (*Figure 6B*, *Figure 7F*, *Figure 7—figure supplement 1A*) compared to Gnai2 neurons indicates a chaperone rich ER environment in these neurons. The chaperone Calr4 has been shown to be enriched in Gnao1 neurons (*Dey and Matsunami, 2011*). Our observations indicate that the enrichment of chaperones is a generalized feature of Gnao1 neurons that extends to several other genes/proteins of related function and localization.

Since it has been proposed that the V2R family is an evolutionarily recent acquisition in rodents and marsupials (*Takigami et al., 2004*) and sub-family members, such as sub-family-C2, A1-A6 V2Rs, as well as H2-M*v* genes have evolved in Muridae (*Young and Trask, 2007*; *Silvotti et al., 2011*), it is possible that mouse Gnao1 neurons may have required to co-evolve ER protein folding mechanisms to handle multiple GPCR/H2-M*v* co-expression and their putative protein interactions. In addition to our observations on the generalized upregulation of ER genes in Gnao1 neurons, it may be possible that some specific ER genes are also necessary for proper folding of vomeronasal type-2 GCPRs. These data and observations assume significance given the well-recognized fact that functional reconstitution of these GPCRs and H2-M*v* molecules in heterologous cells remains a persistent challenge, with some success being reported using the chaperone *Calr4* (*Dey and Matsunami, 2011*). On the other hand, olfactory receptors have been shown to co-opt the ER unfolded protein response (UPR) pathway via the ER proteins PERK/CHOP and transcription factor ATF5 as a mechanism for their functional expression, setting the stage for ONOR expression pattern (*Dalton et al., 2013*). ATF5 is observed in the VNO, however, it is reported to be broadly expressed across Gnai2 and Gnao1 neurons (*Nakano et al., 2016*; *Dalton, 2018b*; *Dalton, 2018a*),. Further experiments are required to evaluate whether V2Rs and Gnao1 neurons adopt similar mechanisms and how the chaperone-rich ER environment observed here might impact the expression, folding of GPCRs, and neuronal function.

It is worth noting that in addition to the expression of ER chaperones in Gnao1 neurons, we also observed distinctly higher levels of ER structural and membrane proteins. Levels of membrane curvature stabilizing proteins - Reep5, NogoB/Reticulon4B, the three-way junction formation protein - Atlastin1 (*Goyal and Blackstone, 2013*), ER sheet lumen spacer protein - Climp-63/Ckap4 as well as Sec61β - a commonly used ER membrane marker were all observed to be high in Gnao1 neurons (*Figure 7G and H*, *Figure 7—figure supplement 1B*). Thus, it is possible that an added layer of complexity could involve modulation of total ER content or ER structure in Gnao1 neuronal subtypes. Our electron microcopy results (*Figure 8*, *Figure 8—figure supplement 1*) revealed a strikingly higher smooth ER membrane content in Gnao1 neurons compared to Gnai2 neurons. Interestingly, this ER adopts a cubic membrane architecture, packing the Gnao1 neuron cell body with a gyroid/sinusoidal membrane. Early electron microscopy studies in rodents, identified that smooth ER content is higher in VSNs than in olfactory sensory neurons (*Ciges et al., 1977*; *Breipohl et al., 1981*; *Taniguchi and Mochizuki, 1982*), however, these studies preceded the discovery of Gnao1-Gnai2 dichotomy. To our knowledge, Breipohl and colleagues (*Breipohl et al., 1981*) reported the presence of smooth ER

whorls in VSNs, later termed as cubic membranes. Future studies using serial section EM would be instrumental in reconstructing the 3D architecture of VSN ER.

Cubic membranes, representing highly curved periodic structures, have been observed in a variety of biological contexts, however, their functional significance has been harder to pin down (*Almsherqi et al., 2006*; *Almsherqi et al., 2009*). In the context of VNO, and in particular Gnao1 neurons, we speculate that the high ER content and dense packing in the form of cubic membranes, could indicate a perpetual stress-like state in these neurons, necessary to address unique folding requirements of V2R or H2-Mv proteins. For instance, the maturation of these proteins and their proper folding or multimerization may require a slower transit through the ER, necessitating the sinusoidal arrangement of ER membranes. Likewise, the V2R biosynthetic processes in Gnao1 neurons may require enhanced levels of ER chaperones, which could in turn induce ER membranes into a homeostatic response. Can V2Rs themselves trigger the upregulation of ER chaperones and the hypertrophic ER, or are these cellular characteristics established before the onset of V2R expression? From our single-cell transcriptomics data, we tried to identify at what stage ER chaperones express during the developmental trajectory of VSNs. We observed that the onset of *Vmn2r1* expression coincides with the upregulation of ER chaperones in cluster n6 immature Gnao1 neurons (*Figure 9A and B*). However, some Gnao1 cells show upregulated ER gene expression without *Vmn2r1/Vmn2r2* expression (*Figure 9B*), hence further experiments are needed to dissect the mechanism and precise relationship between neuronal differentiation, GPCR expression, and ER chaperones as well as ER ultrastructure.

Overall, our data opens a new aspect to look at Gnao1 neurons, when understanding their function: a highly specialized ER environment that is divergent from Gnai2 neurons. The ER genes and their Gnao1 biased expression we identified, could be downstream targets of specific transcription factors and as a result of neuronal differentiation, while functionally these could be required for the proper folding and co-expression of V2R GPCRs. The developmental trajectory of VSNs into Gnai2/Gnao1 neurons via transcription factors and their differential ER environment is summarized as a model in *Figure 9C*.

In conclusion, the comprehensive scRNA seq analysis presented in this study contributes valuable insights into the complexity of the vomeronasal neuroepithelium, offering a roadmap for further investigations into the molecular mechanisms underlying sensory perception and neural development in this specialized sensory organ. Vomeronasal neurons may also serve as a model system to study specialized ER structure-function relationships and their role in the maturation of GPCR-expressing neurons.

# Materials and methods

## Key resources table

| Reagent type (species) or resource | Designation | Source or reference | Identifiers | Additional information |
|---|---|---|---|---|
| Other | Papain | Worthington Biochemical corporation | LK003178 | Enzyme used for single-cell dissociation |
| Other | RNAse-free Deoxyribonuclease I | Roche | 4716728001 | Enzyme used for single-cell dissociation |
| Chemical compound, drug | EDTA | Himedia | ML014-500ML | |
| Other | Ovomucoid inhibitor and albumin mix | Worthington Biochemical corporation | LK003182 | Used in single-cell dissociation |
| Chemical compound, drug | Glycerol | Sigma | G6279-1L | |
| Other | 2 mL Protein Lo-Bind tubes | Eppendorf | 022431102 | Used for single-cell dissociation |
| Other | Pasteur pipettes | Fisher | 13-678-8B | Used in single-cell dissociation |
| Other | HBSS | Hyclone | SH30268.02 | Used in single-cell dissociation |
| Other | EBSS | Hyclone | SH30029.02 | Used in single-cell dissociation |
| Other | 40 um cell strainer | Pluriselect | 43-10040-60 | Used in single-cell dissociation |

*Continued on next page*

*Continued*

| Reagent type (species) or resource | Designation | Source or reference | Identifiers | Additional information |
|---|---|---|---|---|
| Other | 70 um cell strainer | Pluriselect | 43-10070-60 | Used in single-cell dissociation |
| Commercial assay or kit | Chromium Next GEM Single-Cell 3' Kit v3.1, 4rxns | 10 X Genomics | 1000269 | |
| Commercial assay or kit | Chromium Next GEM Chip G Single-Cell Kit | 10 X Genomics | 1000120 | |
| Commercial assay or kit | GoTaq DNA polymerase | Promega | M3005 | |
| Commercial assay or kit | NucleoSpin Gel and PCR Clean-up, Mini kit | Macherey-Nagel | 740609.250 | |
| Antibody | Anti-Digoxigenin-AP (purified Fab fragments from Sheep) | Roche | Cat#11093274910 RRID: AB_514497 | Dilution - 1:7500 |
| Antibody | Anti-Digoxigenin-POD (purified Fab fragments from Sheep) | Roche | Cat#11207733910 RRID: AB_514500 | Dilution - 1:500 |
| Antibody | Anti-Fluorescein-AP (purified Fab fragments from Sheep) | Roche | Cat#11426338910 RRID: AB_2734723 | Dilution - 1:7500 |
| Antibody | Anti-Fluorescein-POD (purified Fab fragments from Sheep) | Roche | Cat#11426346910 RRID: AB_840257 | Dilution - 1:500 |
| Commercial assay or kit | 10 X Flu RNA labeling mix | Roche | 11685619910 | |
| Commercial assay or kit | 10 X Dig RNA labeling mix | Roche | 11277073910 | |
| Peptide, recombinant protein | T7 RNA polymerase | Promega | P2075 | |
| Peptide, recombinant protein | SP6 RNA polymerase | Promega | P1085 | |
| Chemical compound, drug | 5 X Transcription optimized buffer | Promega | P1181 | |
| Peptide, recombinant protein | RNasin Plus | Promega | N2611 | |
| Chemical compound, drug | 100 mM DTT | Promega | P1171 | |
| Chemical compound, drug | Diethyl pyrocarbonate | Sigma | 40718–25 ML | |
| Chemical compound, drug | Akoya Blocking powder | Akoya | FP1020 | |
| Chemical compound, drug | TSA plus Cy3 | Akoya | NEL744001KT | |
| Chemical compound, drug | TSA Plus Fluorescein | Akoya | NEL741001KT | |
| Commercial assay or kit | BCIP/NBT kit | Promega | S3771 | |
| Other | Bovine Serum Albumin | Jackson ImmunoResearch Inc | Cat#001-000-173 RRID:AB_2336947 | Blocking reagent in IHC |
| Other | OCT freezing medium | Leica | Cat#14020108926 | Used for freezing tissue |
| Chemical compound, drug | Paraformaldehyde | Electron Microscopy Sciences | 157–8 | |
| Antibody | anti-Omp (Goat polyclonal) | Wako | Cat#019–22291 RRID:AB_3094987 | Dilution –1:2000 |
| Antibody | anti-Gnao1/GαO (Rabbit polyclonal) | MBL life science | Cat#551 RRID:AB_591430 | Dilution –1:2000 |

*Continued*

| Reagent type (species) or resource | Designation | Source or reference | Identifiers | Additional information |
|---|---|---|---|---|
| Antibody | anti-SEKDEL ER marker (Mouse monoclonal) | Santa Cruz | Cat #sc-58774 RRID:AB_784161 | Dilution –1:1000 |
| Antibody | anti-Grp94 (Rat monoclonal) | Invitrogen | Cat#MA3-016 RRID:AB_2248666 | Dilution –1:1000 |
| Antibody | anti-Hspa5/BiP (Mouse monoclonal) | BD | Cat#610978 RRID:AB_398291 | Dilution –1:1000 |
| Antibody | anti-Atlastin1 (Rabbit polyclonal) | Invitrogen | Cat#PA5-85682 RRID:AB_2792821 | Dilution –1:500 |
| Antibody | anti-PDI (Mouse monoclonal) | Enzo | Cat#ADI-SPA-891 RRID:AB_10615355 | Dilution –1:500 |
| Antibody | anti-Sec61β (Rabbit polyclonal) | Invitrogen | Cat#PA3-015 RRID:AB_2239072 | Dilution –1:400 |
| Antibody | anti-Calnexin (Rabbit polyclonal) | Abcam | Cat#ab10286 RRID:AB_2069009 | Dilution –1:500 |
| Antibody | anti-Reep5 (Rabbit polyclonal) | ProteinTech | Cat#14643–1-AP RRID:AB_2178440 | Dilution –1:500 |
| Antibody | anti-NogoB (Rabbit recombinant monoclonal) | Invitrogen | Cat#MA5-32763 RRID:AB_2810040 | Dilution –1:500 |
| Antibody | anti-Ckap4 (Rabbit polyclonal) | ProteinTech | Cat#16686–1-AP RRID:AB_2276275 | Dilution –1:500 |
| Antibody | anti-Mouse IgG- Alexa Fluor 647 (Donkey polyclonal) | Jackson ImmunoResearch Inc | Cat#715-605-150 RRID:AB_2340862 | Dilution –1:1600 |
| Antibody | Anti-Rat IgG-Cy3 (Donkey polyclonal) | Jackson ImmunoResearch Inc | Cat#712-165-153 RRID:AB_2340667 | Dilution –1:1600 |
| Antibody | Anti-Rabbit IgG-Cy3 (Donkey polyclonal) | Jackson ImmunoResearch Inc | Cat#711-165-152 RRID:AB_2307443 | Dilution –1:1600 |
| Antibody | Anti-Goat IgG-Alexa Fluor 488 (Bovine polyclonal) | Jackson ImmunoResearch Inc | Cat#805-545-180 RRID:AB_2340883 | Dilution –1:1600 |
| Software, algorithm | Cell Ranger v5.0.1 | 10 x genomics; *Zheng et al., 2017* | RRID:SCR_023221 | |
| Software, algorithm | Seurat v4.3.0 | Satija Lab; *Hao et al., 2021* | RRID:SCR_016341 | |
| Software, algorithm | SoupX v1.6.2 | *Young and Behjati, 2020* | RRID:SCR_019193 | https://github.com/constantAmateur/SoupX |
| Software, algorithm | TrimGalore v0.6.6 | TrimGalore | RRID:SCR_011847 | https://github.com/FelixKrueger/TrimGalore/releases/tag/0.6.6 |
| Software, algorithm | Clusterprofiler v4.2.2 | *Wu et al., 2021* | RRID:SCR_016884 | https://bioconductor.org/packages/release/bioc/html/clusterProfiler.html |
| Software, algorithm | Slingshot v2.2.1 | *Street et al., 2018* | RRID:SCR_017012 | https://bioconductor.org/packages/release/bioc/html/slingshot.html |
| Software, algorithm | EnhancedVolcano v1.12.0 | EnhancedVolcano | RRID:SCR_018931 | https://github.com/kevinblighe/EnhancedVolcano |
| Software, algorithm | ggplot2 v3.5.0 | ggplot2 | RRID:SCR_014601 | https://ggplot2.tidyverse.org |
| Software, algorithm | Adobe illustrator CC | Adobe | https://www.adobe.com/products/illustrator | |
| Software, algorithm | ShinyCell | *Ouyang, 2021* | RRID:SCR_022756 | https://github.com/SGDDNB/ShinyCell |

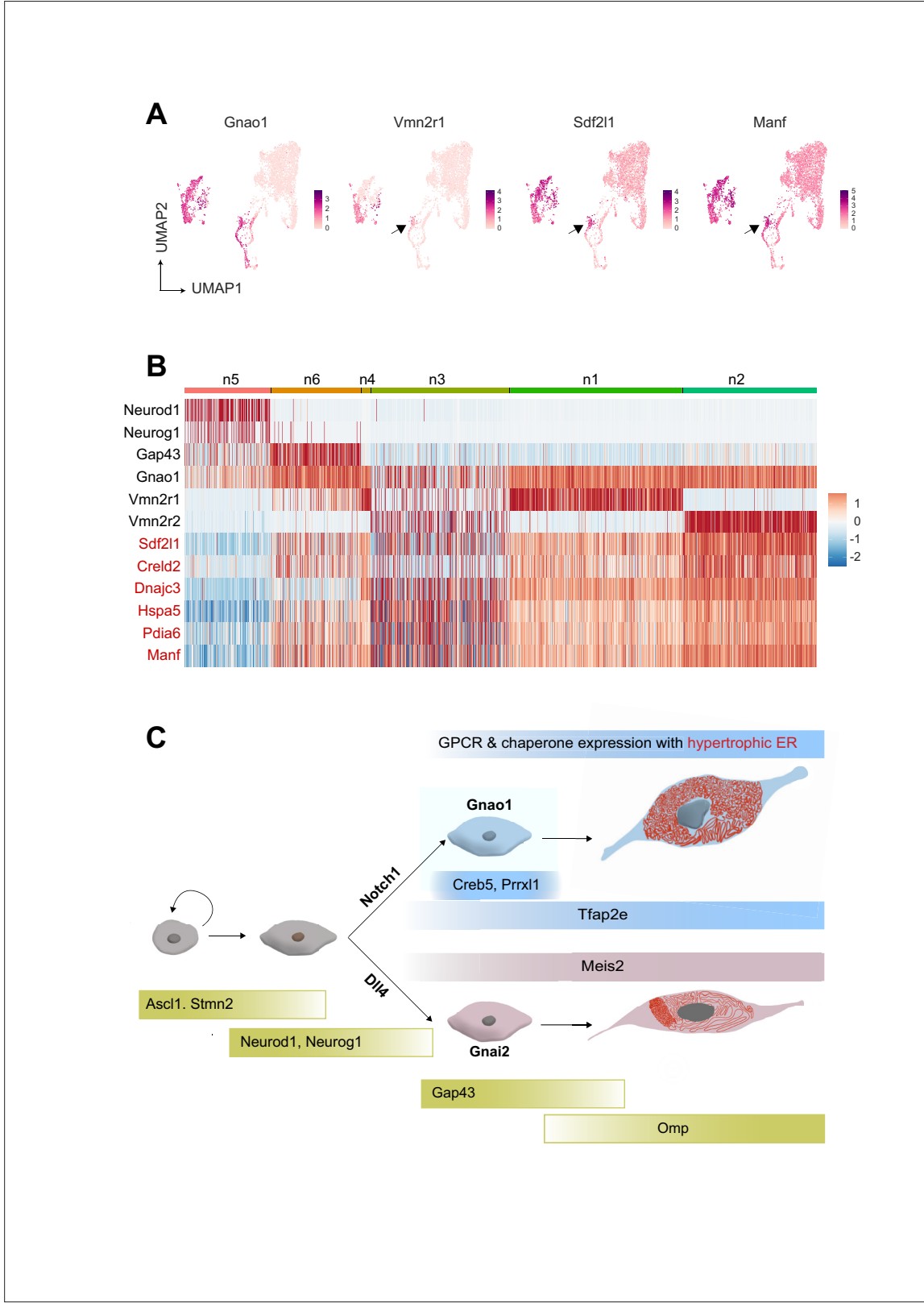

**Figure 9.** Onset of vomeronasal type-2 GPCR (V2R) expression coincides with the expression of endoplasmic reticulum (ER) chaperone genes. (**A**) Feature Plot showing the expression of Gnao1, Vmn2r1, Sdf1l1, and Manf. Sdf2l1 and Manf are known ER chaperones and their upregulation in Gnao1 neurons coincides with Vmn2r1 expression, which is preceded by Gnao1 expression. (**B**) Heatmap showing the expression of Gnao1, Vmn2r1, Vmn2r2, and several ER chaperone genes (red) in the clusters arranged as per their developmental trajectory. (**C**) Cartoon summarizing major transcription factor expression during development leading to Gnao1 neurons with chaperone-rich hypertrophic ER compared to Gnai2 neurons.

## Animals

C57BL/6 J mice were purchased from JAX (RRID:IMSR_JAX:000664) and were used for all experiments. Mice were housed in a specific pathogen-free barrier facility, with a 12 hr light-dark cycle and ad-libitum provision of feed and water. For single-cell RNA sequencing experiments, male and female mice were weaned at postnatal 3 wk age, followed by housing in separate individually ventilated cage racks to avoid exposure to opposite-sex stimuli, and used at 7–8 wk age. All experiments were carried out with approval from the Institutional Animal Ethics Committee of TIFR Hyderabad (Protocol number TCIS/2019/05).

## VNO dissociation

Papain dissociation buffer (PDB) was made by reconstituting a single use papain vial from Worthington Biochemical Corporation with 5 mL Earle's balanced salt solution (EBSS) (pH 7.2) and warmed to 37 °C until the solution appeared clear by maintaining 95% $CO_2$, 5% $O_2$ environment. VNOs were dissected from male and female animals (six male, 10 female) and processed separately. The sensory epithelium was separated using forceps and immediately placed in EBSS equilibrated with 95% $CO_2$, and 5% $O_2$. 60 U of DNase was added to the prewarmed PDB made earlier and 3 mL of it was transferred to a single well of 12-well dish. Neuroepithelial tissue from multiple animals was transferred to this well containing the final dissociation buffer and was cut into small pieces, and the suspension was transferred to a 14 mL tube and was incubated with gentle shaking at 37 °C for 30 min on a thermomixer (Eppendorf) by passing 95% $CO_2$, 5% $O_2$ on top of the liquid headspace. During this incubation, the solution was triturated with a fire-polished Pasteur pipette every 10 min and at the end of the incubation. After incubation, the suspension was passed sequentially through 70 μm, and 35 μm filters to remove tissue debris. The filtered suspension was further layered over a mix of ovomucoid inhibitor (2.5 mg/mL) and albumin (2.5 mg/mL) in EBSS and centrifuged at 400 × g for 5 min at room temperature to remove subcellular debris or membrane fragments. The supernatant was discarded, and the cell pellet was used for the next steps after Hank's balanced Salt solution (HBSS) wash.

## Single-cell library preparation and sequencing

Dissociated neurons resuspended in HBSS were used for library preparation. Using a hemocytometer and trypan blue staining, cell density was estimated at 1200 cells/μl and 1100 cells/μl for male and female samples, respectively with greater than 80% cell viability. Single-cell capture, and library preparation were done using a Chromium Next GEM Single-Cell 3' GEM, Library and Gel Bead Kit v3.1 on a 10 X Chromium controller (10 X Genomics). The volume of suspension to be loaded was decided as per the manufacturer's recommendation to ensure the target capture of 6000 cells per well. Single-cell suspension from male and female samples was loaded onto two separate wells of different chips giving a total of four libraries (two male and two female samples). Each library was barcoded and sequenced separately on a single lane of HiSeq X to obtain a mean depth of at least 100,000 reads per cell in a 2 × 150 bp configuration.

## Single-cell RNA sequencing data analysis

### Trimming the reads

To obtain an informative portion of raw reads, they were hard trimmed to make sure read-1 and read-2 are 28 and 91 bps long as per requirements specified by 10 X genomics using *trim_galore*.

### Alignment, UMI counting, and cell calling

Mouse reference genome mm10 (Mus_musculus.GRCm38.dna.primary_assembly.fa) and GTF file were downloaded from Ensembl (Genome build GRCm38.p6). The GTF file was filtered to retain only protein-coding transcripts by removing readthrough and any non-coding transcripts using *cell ranger mkgtf* using attribute = gene_biotype:protein_coding and readthrough_transcript. A custom reference was built with *cell ranger mkref* using filtered GTF and mm10 genome. Alignment to custom reference, UMI counting, and cell calling, removal of empty droplets, and count matrix generation were done in a single step using *cell ranger count* individually for each sample using default parameters.

## Integrating samples and read depth normalization

After alignment, integration of two male and two female samples was done at raw data level ensuring an average number of confidently mapped reads for each sample is equal using *cell ranger aggr* pipeline. This pipeline subsampled reads in higher-depth samples to ensure the sequencing depth is normalized across samples. During this step, a sample suffix (1–4) was added to each cell barcode to distinguish its source and to avoid barcode clashes in the integrated count matrix.

## Ambient RNA correction, quality check, and filtering

Before downstream analysis stringent filtering was implemented to retain high-quality cells. Ambient RNA contamination was removed using the soupX package with default parameters in auto mode. The adjusted count matrix from soupX consisting of 10,615 cells was used as an input to the Seurat package using the *CreateSeuratObject* function. To remove potential doublets and low gene count cells, an additional filter was applied to select cells that express 200–7000 genes resulting in dropping 683 cells.

## Normalization, scaling, dimensionality reduction, clustering, marker identification, and additional filtration

The data was normalized using the *NormalizeData* function using the LogNormalize method that log-transforms the expression counts after multiplying with scaling factor 10000. After normalization, the top 2000 variable genes in the dataset were identified using the *FindVariableFeatures* function using mean.function=ExpMean, dispersion.function=LogVMR, x.low.cutoff=0.0125, x.high.cutoff=3, y.cutoff=0.5 as parameters. This variable gene set was used for all downstream analysis including dimensionality reduction and clustering. All vomeronasal receptors were amongst the variable features gene set. Data was scaled and centered using *ScaleData* using the default model by regressing out a percentage of mitochondrial genes and the total number of RNA molecules per cell to remove their contribution in downstream analysis. The basis of scaling for each gene is by subtracting the mean from the value and dividing the difference by the standard deviation. To cluster the cells, we initially performed principal component analysis (PCA) with 50 components using *RunPCA*. We used elbow plot and Jack Straw plot and determined the optimum number of dimensions required for the next steps of clustering as 37. Graph-based clustering was performed using Seurat's *FindNeighbours* function to identify the neighboring cells that share similar expression patterns in a network constructed in PCA space with 37 dimensions. Later the clusters were identified by using the *FindClusters* function by varying the resolution parameter from 0.2 to 0.8. The resolution parameter 0.3 was chosen, as it shows minimum overlapping markers upon plotting the heatmap of gene markers for each cluster identified by the *FindAllMarkers* function. To clean up the dataset further, cells (408) expressing the Hbb-bs gene were considered as RBC contamination, and a cluster (215 cells) enriched with mitochondrial genes indicative of dying cells was removed from the data. This resulted in a total of 9180 cells.

## Dimensionality reduction for 2D representation and cell type assignment

UMAP was generated using the *RunUMAP* function. Cluster identity was assigned based on known markers of each cell type. Two clusters expressing Gnai2 as a major marker were merged as they had very similar expression profiles. Solitary chemosensory cells and endothelial cells were manually assigned a cluster identity based on the expression of Trpm5/Rgs21 and Aqp1/Egfl7 and by selecting the cells using the *CellSelector* function of Seurat.

## Comparison of male and female data

Based on suffix assigned to barcodes during the integration of male and female samples, a metadata column was added to the Seurat object marking the sex of source tissue as male or female. Furthermore, cells in each cluster were divided by appending the sex to the cluster identities. Differential expression analysis was performed between cells from male and female VNO for each cluster using the *FindMarkers* function of Seurat.

## Clustering and downstream analysis of neurons

For neuron-specific analysis, neuronal cell types representing Gnao1 neurons, Gnai2 neurons, immature neurons, and progenitor cells were separated from the main Seurat object using *subset* function

to create a new 'neurons' object. The data was scaled again, PCA was performed, the number of principal components was determined, and clustering was performed as described above with resolution = 0.4 to define neuronal subtypes. To compare the results of clustering with and without VRs, a new Seurat object– 'neurons_no_VR' was created by removing all vomeronasal receptors from the variable gene set. Dimensionality reduction, and clustering were performed again using the same method mentioned above. Seurat's query to reference mapping module was used to project the neurons_no_VR object in the same UMAP space as the neurons object so that UMAPs are comparable. The anchors were identified by using the *FindTransferAnchors* function with neurons object as reference and neurons_no_VR as query. The reference UMAP model was computed using the *MapQuery* function using the anchors.

## Pseudotime analysis
Developmental trajectory of VSNs was inferred using the SlingShot package. 'neurons' objects with five major clusters: progenitor cells, Gnao1+ immature neurons, Gnai2+ immature neurons, mature Gnao1 neurons, and mature Gnai2 neurons. Progenitor cells/cluster n5 was chosen as the starting cluster.

## Differential gene expression of Gnao1, Gnai2 neurons
Differentially expressed genes between Gnao1 and Gnai2 neurons (mature or immature) were identified using the *FindMarkers* function of Seurat on the 'neurons' object with default parameters. The results are plotted as volcano plots using the EnhancedVolcano package. Non-parametric Wilcoxon rank sum test was used for calculating p-values and the Bonferroni method to calculate adjusted p-values for multiple testing.

## GO analysis
Gene ontology analysis of Gnao1 enriched genes was done with cluster Profiler package using *enricher* function on mouse gene sets related to mouse biological processes (GO:BP) ontology downloaded from Mouse Molecular Signatures Database. The enrichment analysis was restricted to the differentially expressed genes with log2 fold change greater than 1.

## Co-expression analysis
Non-zero expression level for a particular gene may not indicate that the VR is expressed. Therefore, we identified the cut-off for V1R, V2R, and H2-Mvs based on the distribution of expression levels across all cells. Normalized gene expression values of V1R, V2R, and H2-Mv genes were extracted from each cell of Gnai2 and Gnao1 clusters and the distribution is plotted across all cells. This resulted in a Bi-modal distribution and starting of the second peak was considered as a cut-off for V2R/V1R/H2-Mv genes (*Figure 4—figure supplement 1D and F*). The genes were considered co-expressed in a single-cell if their normalized expression value was greater than the identified threshold value of 1.25 for V2R, H2-Mv gene, and 2.5 for V1R genes. Furthermore, the number of cells crossing the threshold was calculated for each combination.

Raw single-cell RNA seq data of VNO from p56 animals generated by *Hills et al., 2024* was downloaded from NCBI GEO with ID: GSE252365. The data was analyzed in the same method as described above and Gnao1 neurons from mature neuronal clusters identified based on the expression of Gnao1, Omp were used for co-expression analysis.

The graphical user interface for scvnoexplorer.com was made using the ShinyCell tool *Ouyang, 2021*. Reference details of software used for data analysis are mentioned in a tabular form below.

## RNA in situ hybridization (ISH)
### Probe design and synthesis
Primers (*Supplementary file 8*) targeting unique regions of each gene were designed by adding a T7/SP6 promoter sequence. PCR (GoTaq DNA polymerase) was performed using gene specific primers with VNO cDNA as template and the product was run on an agarose gel to confirm the specific amplification of each gene. When multiple bands were seen, the band with desired molecular weight was

cut from the gel, and PCR product clean-up or gel purification was performed using NucleoSpin Gel and PCR Cleanup kit. Purified PCR product was verified by Sanger sequencing and was used as the template for in vitro transcription to obtain digoxigenin (Dig) and fluorescein (Flu) labeled riboprobes using the following reaction composition: 1 ug Template DNA, 1 x Dig/Flu RNA labeling mix, 1 U/uL RNAsin plus, 5 mM DTT, 1 x Transcription buffer in nuclease-free water. The reaction was purified using a Qiagen/MN RNA cleanup kit and stored at –80°C by adding formamide to 50% of the volume.

H2-Mv genes and Gnai2 were cloned to a plasmid vector with T7 or SP6 promoters. Gnao1 plasmid was purchased from Invitrogen (6309166). The plasmid was linearized, and in vitro transcription was performed as described earlier.

## Chromogenic ISH

Fresh VNO was embedded in OCT and 14 μm thick sections were collected on a cryostat. Tissue sections were fixed with 4% PFA in Diethyl Pyrocarbonate (DEPC) treated phosphate buffer saline (PBS) and acetylation was performed using Propionic Anhydride, Triethanolamine, and NaCl. Permeabilization was done using 0.1 M HCl and sections were pre-hybridized using hybridization buffer (50% Formamide, 5 X SSC, 5 X Denhardt's solution, 0.1 mg/mL Salmon sperm DNA, 0.25 mg/mL yeast t-RNA) for 2 hr. Probes were diluted (1:100) in the hybridization buffer and added to the slides. Hybridization was performed for 12–16 hr at 67 °C. To remove the unbound probe, sections were washed thrice with 0.2 X SSC for 30 min each. Slides were equilibrated in a buffer consisting of 0.1 M Tris-HCl. pH 7.5, 150 mM NaCl for 5 min and blocked with 10% FBS in the same buffer. Hybridized Digoxigenin (DIG) or Fluorescein (Flu) containing RNA probes were detected by alkaline phosphatase-conjugated Anti-DIG or Anti-Flu Fab$_2$ fragments (1:7500 dilution). Unbound antibody was washed, and the development of alkaline phosphatase was done using BCIP/NBT substrate diluted as per manufacturer's protocol in 0.1 M Tris-HCl pH–9, 0.1 M NaCl, 50 mM MgCl$_2$. The signal was developed for 12–72 hr based on the intensity. After development, slides were washed with PBS and mounted using 10% glycerol in 0.1 M Tris-HCl (pH 7.5). ISH for each gene was done twice with biological replicates.

## Two-color fluorescence ISH

As described above, the same protocol for chromogenic ISH was followed until 0.2 X SSC washes, after which the hybridized probes were detected using peroxidase-conjugated anti-FLU or anti-DIG Fab(2) fragments and the Tyramide signal amplification (TSA) system from Akoya Biosciences. Slides were incubated in 3% H$_2$O$_2$ in PBS for 1 hr and washed thrice for 10 min each and blocked with 0.5% blocking buffer for 30 min. Dig and Flu were sequentially developed using TSA-FITC or TSA-Cy3.

## Immunohistochemistry

VNOs were dissected from animals of age 8–12 wk, fixed with 4% Paraformaldehyde (PFA) in PBS, and cryopreserved with 30% sucrose. Tissue was embedded in an OCT-freezing medium and cryosections of 14 μm thickness were collected on glass slides. Sections were post-fixed again with 4% PFA in PBS, blocked, and permeabilized by incubating for 2 hr with a blocking buffer (3% Bovine Serum Albumin, 0.1% TritonX-100 in PBS with 0.02% Sodium Azide). After blocking, the sections were incubated with primary antibodies diluted in the blocking buffer for 2 hr followed by washing off excess antibody and secondary antibody incubation for 2 more hours. All washes were done using 0.1% TritonX-100 in PBS.

## Light microscopy image acquisition

Chromogenic images were acquired on an Olympus BX43 upright microscope using bright field Koehler illumination and 10 X objective (PlaN, 0.24 NA) equipped with an Olympus DP25 color CCD camera. Two- or three-color fluorescent RNA-ISH images and fluorescent Immunohistochemistry images were acquired sequentially using Leica Stellaris upright confocal microscope using 10 X (HC PL APO 0.40 NA) or 63 X (HC PL APO 1.40–0.60 NA Oil) objective keeping 1 Airy unit as pinhole size. Acquisition parameters were adjusted and calibrated using single-channel controls to ensure that there was no spectral cross-talk between channels in multi-color imaging experiments. Immunohistochemistry images of ER antibodies were pseudo-colored using the ImageJ Fire LUT with 0–255 linear scaling.

## Quantification of ER antibody signal intensities

VNO sections were labeled with a single ER antibody, along with goat anti-OMP and rabbit anti-Gnao1 or mouse anti-KDEL as markers of neurons and Gnao1 zone, respectively. At least 20 sections from three biological replicates were imaged via confocal microscopy and used for each ER antibody quantification. 50-pixel wide rectangular regions of interest (ROI) were drawn along the apical to basal axis and the signal intensity along the ROI for each channel was extracted from the images using Fiji software. To make the signal intensities along the apical-basal axis across different images and antibodies comparable, the signal intensity and length of ROI were normalized by min-max normalization,

$$x' = \frac{x - \min(x)}{\max(x) - \min(x)}$$

where, $x$ is the measured or actual value, $\min(x)$ and $\max(x)$ are minimum and maximum values of $x$, respectively. The trendline of normalized signal intensity along the ab axis was generated by fitting a smoothened curve using the *geom_smooth* function of the ggplots package based on the generalized additive model for each antibody.

## Electron microscopy

VNOs were dissected from animals that were trans-cardially perfused with PBS followed by 4% PFA in PBS buffer. Dissected VNO was embedded in 2% agarose and coronal sections of 400 um were obtained on a vibratome in 0.1 M sodium cacodylate buffer (pH 7.4). Sections were fixed with Karnovsky's fixative (3% PFA + 2% glutaraldehyde in 0.1 M sodium cacodylate) for up to 1 wkk. Subsequent processing steps were similar to those described before (*Terasaki et al., 2020*). Briefly, vibratome sections were rinsed with 0.1 M sodium cacodylate and then immersed in 1% osmium, and 0.8% potassium ferricyanide in the cacodylate buffer for 1 hr. This was followed by incubation in 1% aqueous uranyl acetate for 1 hr, then 30 min in lead aspartate (*Walton, 1979*), dehydration in graded ethanol, infiltration with epon resin then embedding and polymerization at 60 °C for 2 d. Serial 70 nm thick sections were collected using a Powertome automated tape collector (RMC Boeckeler, Tucson, AZ). The tape was mounted on a silicon wafer, carbon coated, and imaged with a field emission electron microscope (Thermo Fisher/FEI Verios, Hillsboro, OR). Backscatter electrons were imaged from a 5 keV/0.8 nanoAmp beam of electrons.

## Acknowledgements

We acknowledge Jyoti Rohilla, and Nandana Nanda for their help in the preparation of DNA templates for riboprobe generation, Tulasi Nagabandi for helping with scRNA library preparation, Tamal Das for sharing Atlastin1, Sec61β antibodies and Erik Snapp for insightful discussions. We acknowledge support from the Department of Atomic Energy, Government of India, under Project No. RTI 4007.

## Additional information

### Funding

| Funder | Grant reference number | Author |
| --- | --- | --- |
| Department of Atomic Energy, Government of India | Project No. RTI 4007 | Adish Dani |

The funders had no role in study design, data collection and interpretation, or the decision to submit the work for publication.

### Author contributions

GVS Devakinandan, Conceptualization, Data curation, Software, Formal analysis, Investigation, Visualization, Methodology, Writing – original draft, Writing – review and editing; Mark Terasaki, Investigation, Visualization, Writing – review and editing, Electron microscopy and analysis; Adish Dani,

Conceptualization, Resources, Data curation, Supervision, Funding acquisition, Investigation, Visualization, Methodology, Writing – original draft, Project administration, Writing – review and editing

**Author ORCIDs**
GVS Devakinandan https://orcid.org/0009-0009-4978-7477
Mark Terasaki https://orcid.org/0000-0003-4964-9401
Adish Dani https://orcid.org/0000-0002-5491-7709

**Ethics**

Animal handling and experiments were carried out with approval from Institutional Animal Ethics Committee of TIFR Hyderabad (Protocol number TCIS/2019/05).

Reviewer #1 (Public review): https://doi.org/10.7554/eLife.98250.3.sa1
Reviewer #2 (Public review): https://doi.org/10.7554/eLife.98250.3.sa2
Reviewer #3 (Public review): https://doi.org/10.7554/eLife.98250.3.sa3
Author response https://doi.org/10.7554/eLife.98250.3.sa4

## Additional files

**Supplementary files**

• Supplementary file 1. List of top 20 genes specific to each cluster corresponding to *Figure 1A and B*.

• Supplementary file 2. Frequency of cells belonging to various clusters corresponding to *Figure 1A*.

• Supplementary file 3. List of top 30 highly expressed genes in three macrophage clusters (corresponding to *Figure 2B and C*) selected based on log2Fold change comparison of each cluster with the other two clusters.

• Supplementary file 4. A list of the top 30 markers of neuronal clusters (n1-n13) from *Figure 3—figure supplement 1*, identified based on average log2FC after applying filters: adjusted p-value <0.005, fraction of cells expressing the gene in the cluster >=0.5 and fraction of cells expressing the gene other clusters <0.5.

• Supplementary file 5. Frequency of co-expression of vomeronasal type-2 GPCRs (V2Rs) and H2-Mv genes in Gnao1 neurons (corresponding to *Figure 4*). The frequency of family C, ABD V2Rs, H2-Mv, V2Rs-H2-Mv co-expression combinations are separated into sheets.

• Supplementary file 6. Frequency of expression of vomeronasal type-I GPCRs (V1R) combinations in mature Gnai2 neurons, corresponding to *Figure 4—figure supplement 4*.

• Supplementary file 7. Gene ontology terms (biological processes category) associated with genes for which RNA in situ hybridization was performed in *Figure 6* and *Figure 5—figure supplement 1*.

• Supplementary file 8. Primer sequences used for PCR amplification of gene-specific DNA template to generate riboprobes for in situ hybridization (ISH)col.

• MDAR checklist

**Data availability**

Single cell RNA sequencing data was deposited to GEO under accession ID GSE253252. Visualization of gene expression patterns from this study are made publicly available at https://www.scvnoexplorer.com.

The following dataset was generated:

| Author(s) | Year | Dataset title | Dataset URL | Database and Identifier |
|---|---|---|---|---|
| Devakinandan GVS, Dani A | 2024 | Single-cell transcriptomics of vomeronasal neuroepithelium reveals a differential endoplasmic reticulum environment amongst neuronal subtypes | https://www.ncbi.nlm.nih.gov/geo/query/acc.cgi?acc=GSE253252 | NCBI Gene Expression Omnibus, GSE253252 |

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
