## [Editor Report · eLife Assessment]

This is a **valuable** manuscript analyzing single-cell RNA-sequencing data from the mouse vomeronasal organ. **Convincing** evidence in this manuscript allows the authors to identify and verify the differential expression of genes that distinguish apical and basal vomeronasal neurons. The authors also show that Gnao1 neurons exhibit enriched expression of ER-related genes, which they verify with in situ hybridizations and immunostaining and also explore via electron microscopy.

---

## [Referee Report · Reviewer #1 (Public review)]

Devakinandan et al. present a revised version of their manuscript. Their scRNA-seq data is a valuable resource to the community, and they further validate their findings via in situ hybridizations and electron microscopy. Overall, they have addressed my major concerns. I only have two minor comments.

(1) The authors note in Figure 4I, and K that because the number of C2 V2Rs or H2-Mv receptors increased while the normalized expression of Gnao1 remained constant (and likewise for V1Rs and Gnai2 in Figure 4-S4C) that their results are unlikely to be capturing doublets. I'm not sure that this is the case. If the authors added together two V2R cells the total count of every gene might double, but the normalized expression of Gnao1 would remain the same. To address this concern, the authors should also show the raw counts for Gnao1 as well as the total number of UMIs for these cells.

(2) As requested, the authors have now added a colorbar to the pseudocolored images in Figures 7. However, this colorbar still doesn't have any units. Can the authors add some units, or clarify in the methods how the raw data relates to the colors (e.g. is it mapped linearly, at a logscale, with gamma or other adjustments, etc.)? Moreover, it's also unclear what the dots in the backgrounds of plots like Figure 7E mean. Are they pixels? Showing the individual lines, the average for each animal, or omitting them entirely, might make more sense.

---

## [Referee Report · Reviewer #2 (Public review)]

Summary:

The study focuses on the vomeronasal organ, the peripheral chemosensory organ of the accessory olfactory system, by employing single-cell transcriptomics. The author analyzed the mouse vomeronasal organ, identifying diverse cell types through their unique gene expression patterns. Developmental gene expression analysis revealed that two classes of sensory neurons diverge in their maturation from common progenitors, marked by specific transient and persistent transcription factors. A comparative study between major neuronal subtypes, which differ in their G-protein sensory receptor families and G-protein subunits (Gnai2 and Gnao1, respectively), highlighted a higher expression of endoplasmic reticulum (ER) associated genes in Gnao1 neurons. Moreover, distinct differences in ER content and ultrastructure suggest some intriguing roles of ER in Gnao1-positive vomeronasal neurons. This work is likely to provide useful data for the community and is conceptually novel with the unique role of ER in a subset of vomeronasal neurons.

Strengths:

(1) The study identified diverse cell types based on unique gene expression patterns, using single-cell transcriptomic.

(2) The analysis suggest that two classes of sensory neurons diverge during maturation from common progenitors, characterized by specific transient and persistent transcription factors.

(3) A comparative study highlighted differences in Gnai2- and Gnao1-positive sensory neurons.

(4) Higher expression of endoplasmic reticulum (ER) associated genes in Gnao1 neurons.

(5) Distinct differences in ER content and ultrastructure suggest unique roles of ER in Gnao1-positive vomeronasal neurons.

(6) The research provides conceptually novel on the unique role of ER in a subset of vomeronasal neurons, offering valuable insights to the community.

Comments on latest version:

In the revised manuscript, the authors have thoroughly addressed all of this reviewer's concerns.

---

## [Referee Report · Reviewer #3 (Public review)]

Summary:

In this manuscript, Devakinandan and colleagues have undertaken a thorough characterization of the cell types of the mouse vomeronasal organ, focusing on the vomeronasal sensory neurons (VSNs). VSNs are known to arise from a common pool of progenitors that differentiate into two distinct populations characterized by the expression of either the G protein subunit Gnao1 or Gnai2. Using single-cell RNA sequencing followed by unsupervised clustering of the transcriptome data, the authors identified three Gnai2+ VSN subtypes and a single Gnao1+ VSN type. To study VSN developmental trajectories, Devakinandan and colleagues took advantage of the constant renewal of the neuronal VSN pool, which allowed them to harvest all maturation states. All neurons were re-clustered and a pseudotime analysis was performed. The analysis revealed the emergence of two pools of Gap43+ clusters from a common lineage, which differentiate into many subclusters of mature Gnao1+ and Gnai2+ VSNs. By comparing the transcriptomes of these two pools of immature VSNs, the authors identified a number of differentially expressed transcription factors in addition to known markers. Next, by comparing the transcriptomes of mature Gnao1+ and Gnai2+ VSNs, the authors report an enrichment of ER-related genes in Gnao1+ VSNs. Using electron microscopy, they found that this enrichment was associated with specific ER morphology in Gnao1+ neurons. Finally, the authors characterized chemosensory receptor expression and co-expression (as well as H2-Mv proteins) in mature VSNs, which recapitulated known patterns.

Strengths:

The data presented here provide new and interesting perspectives on the distinguishing features between Gnao1+ and Gnai2+ VSNs. These features include newly identified markers, such as transcription factors, as well as an unsuspected ER-related peculiarity in Gnao1+ neurons, consisting in a hypertrophic ER and an enrichment in ER-related genes. In addition, the authors provide a comprehensive picture of specific co-expression patterns of V2R chemoreceptors and H2-Mv genes.

Importantly, the authors provide a browser (scVNOexplorer) for anyone to explore the data, including gene expression and co-expression, number and proportion of cells, with a variety of graphical tools (violin plots, feature plots, dot plots, ...).

---

## [Author Response]

The following is the authors’ response to the current reviews.

**Reviewer #1 (Public review):**
Devakinandan et al. present a revised version of their manuscript. Their scRNA-seq data is a valuable resource to the community, and they further validate their findings via in situ hybridizations and electron microscopy. Overall, they have addressed my major concerns. I only have two minor comments.(1) The authors note in Figure 4I, and K that because the number of C2 V2Rs or H2-Mv receptors increased while the normalized expression of Gnao1 remained constant (and likewise for V1Rs and Gnai2 in Figure 4-S4C) that their results are unlikely to be capturing doublets. I'm not sure that this is the case. If the authors added together two V2R cells the total count of every gene might double, but the normalized expression of Gnao1 would remain the same. To address this concern, the authors should also show the raw counts for Gnao1 as well as the total number of UMIs for these cells.

In Figure 4I, 4K and Figure 4-Figure supplement 4C, on Y-axis, we plotted the sum of normalized counts of all V1R/V2R/H2-Mv genes expressed in each cell along with the normalized expression value of Gnao1/Gnai2. Both VR/H2-Mv and Gnao1/Gnai2 are normalized values, with normalization based on LogNormalize (mentioned in methods). We show here plots of total expression calculated from raw counts corresponding to the same Figure. Raw counts of VRs/H2-Mv, Gnao1/Gnai2 are plotted separately due to difference in scale. The overall trend matches normalized counts, with minor fluctuations in Gnao1/Gnai2.

**Author response image 1. sa4fig1:** 

As mentioned in our response to version-1 reviews and in our manuscript, doublets generally are a random combination of two cells and the probability that a combinatorial pattern is due to doublet is proportional to the abundance of cells expressing those genes. It is possible that some of the family-C V2R combinations represented by 2 cells are doublets because of their widespread expression. The frequency of combinatorial expression patterns, greater than a set threshold of 2 cells, that we observed for family ABD V2Rs or V1Rs (supplementary tables 7, 8) is an indication of co-expression and unlikely from random doublets. For instance, 134 cells express two V1Rs, of which 44 cells express Vmn1r85+Vmn1r86, 21 cells express Vmn1r184+Vmn1r185, 13 express Vmn1r56+Vmn1r57, 6 express Vmn1r168+Vmn1r177. Some of the co-expression combinations we reported were also identified and verified experimentally in Lee et al., 2019 and Hills et. al., 2024.

The co-expression of multiple family-C2 V2Rs (Vmn2r2-Vmn2r7) along with ABD V2Rs per cell as shown in our data, has been shown experimentally in earlier studies.

(2) As requested, the authors have now added a colorbar to the pseudocolored images in Figures 7. However, this colorbar still doesn't have any units. Can the authors add some units, or clarify in the methods how the raw data relates to the colors (e.g. is it mapped linearly, at a logscale, with gamma or other adjustments, etc.)? Moreover, it's also unclear what the dots in the backgrounds of plots like Figure 7E mean. Are they pixels? Showing the individual lines, the average for each animal, or omitting them entirely, might make more sense.

We used the Fire LUT with linear scale within Fiji / Image-J software to assign scale to the pseudo-colored images in Figure 7. We will include this description in our methods and thank the reviewer for pointing it out. The dots in the background are mentioned in Figure 7 legend as fluorescence intensity values normalized to a 0-1 scale and color coded for each antibody. The trendline was fitted on these values.

**Reviewer #2 (Public review):**
Summary:The study focuses on the vomeronasal organ, the peripheral chemosensory organ of the accessory olfactory system, by employing single-cell transcriptomics. The author analyzed the mouse vomeronasal organ, identifying diverse cell types through their unique gene expression patterns. Developmental gene expression analysis revealed that two classes of sensory neurons diverge in their maturation from common progenitors, marked by specific transient and persistent transcription factors. A comparative study between major neuronal subtypes, which differ in their G-protein sensory receptor families and G-protein subunits (Gnai2 and Gnao1, respectively), highlighted a higher expression of endoplasmic reticulum (ER) associated genes in Gnao1 neurons. Moreover, distinct differences in ER content and ultrastructure suggest some intriguing roles of ER in Gnao1-positive vomeronasal neurons. This work is likely to provide useful data for the community and is conceptually novel with the unique role of ER in a subset of vomeronasal neurons.Strengths:(1) The study identified diverse cell types based on unique gene expression patterns, using single-cell transcriptomic.(2) The analysis suggest that two classes of sensory neurons diverge during maturation from common progenitors, characterized by specific transient and persistent transcription factors.(3) A comparative study highlighted differences in Gnai2- and Gnao1-positive sensory neurons.(4) Higher expression of endoplasmic reticulum (ER) associated genes in Gnao1 neurons.(5) Distinct differences in ER content and ultrastructure suggest unique roles of ER in Gnao1-positive vomeronasal neurons.(6) The research provides conceptually novel on the unique role of ER in a subset of vomeronasal neurons, offering valuable insights to the community.
**Reviewer #3 (Public review):**
Summary:In this manuscript, Devakinandan and colleagues have undertaken a thorough characterization of the cell types of the mouse vomeronasal organ, focusing on the vomeronasal sensory neurons (VSNs). VSNs are known to arise from a common pool of progenitors that differentiate into two distinct populations characterized by the expression of either the G protein subunit Gnao1 or Gnai2. Using single-cell RNA sequencing followed by unsupervised clustering of the transcriptome data, the authors identified three Gnai2+ VSN subtypes and a single Gnao1+ VSN type. To study VSN developmental trajectories, Devakinandan and colleagues took advantage of the constant renewal of the neuronal VSN pool, which allowed them to harvest all maturation states. All neurons were re-clustered and a pseudotime analysis was performed. The analysis revealed the emergence of two pools of Gap43+ clusters from a common lineage, which differentiate into many subclusters of mature Gnao1+ and Gnai2+ VSNs. By comparing the transcriptomes of these two pools of immature VSNs, the authors identified a number of differentially expressed transcription factors in addition to known markers. Next, by comparing the transcriptomes of mature Gnao1+ and Gnai2+ VSNs, the authors report an enrichment of ER-related genes in Gnao1+ VSNs. Using electron microscopy, they found that this enrichment was associated with specific ER morphology in Gnao1+ neurons. Finally, the authors characterized chemosensory receptor expression and co-expression (as well as H2-Mv proteins) in mature VSNs, which recapitulated known patterns.Strengths:The data presented here provide new and interesting perspectives on the distinguishing features between Gnao1+ and Gnai2+ VSNs. These features include newly identified markers, such as transcription factors, as well as an unsuspected ER-related peculiarity in Gnao1+ neurons, consisting in a hypertrophic ER and an enrichment in ER-related genes. In addition, the authors provide a comprehensive picture of specific co-expression patterns of V2R chemoreceptors and H2-Mv genes.Importantly, the authors provide a browser (scVNOexplorer) for anyone to explore the data, including gene expression and co-expression, number and proportion of cells, with a variety of graphical tools (violin plots, feature plots, dot plots, ...).

The following is the authors’ response to the original reviews.

**Public Reviews:**

**Reviewer #1 (Public Review):**
Devakinandan and colleagues present a manuscript analyzing single-cell RNAsequencing data from the mouse vomeronasal organ. The main advances in this manuscript are to identify and verify the differential expression of genes that distinguish apical and basal vomeronasal neurons. The authors also identify the enriched expression of ER-related genes in Gnao1 neurons, which they verify with in situ hybridizations and immunostaining, and also explore via electron microscopy. Finally, the results of this manuscript are presented in an online R shiny app. Overall, these data are a useful resource to the community. I have a few concerns about the manuscript, which I've listed below.General Concerns:(1) The authors mention that they were unable to identify the cells in cluster 13. This cluster looks similar to the "secretory VSN" subtype described in a recent preprint from C. Ron Yu's lab (10.1101/2024.02.22.581574). The authors could try comparing or integrating their data with this dataset (or that in Katreddi et al. 2022) to see if this is a common cell type across datasets (or arises from a specific type of cell doublets). In situ hybridizations for some of the marker genes for this cluster could also highlight where in the VNO these cells reside.

Cluster13 (Obp2a+) cells identified in our study have similar gene expression markers to “putative secretory” cells mentioned in Hills et al.. At the time this manuscript was available publicly, our publication was already communicated. We have now performed RNA-ISH to Obp2a, the topmost marker identified with this cluster, and found it to be expressed in cells from glandular tissue on the non-sensory side. Some of the other markers associated with this cluster such as Obp2b, Lcn3, belong to the lipocalin family of proteins. Hence in our estimate these markers collectively represent non-sensory glandular tissue. We have added Obp2a RNA-ISH to Figure 2-figure supplement-1A and results section in our revised manuscript. Cluster-13 also has cells expressing Vmn1r37, which typically is expressed in neuronal cells. However, we do not see Obp2a mRNA in the sensory epithelium. It is possible that cluster-13 comprises a heterogenous mixture of cells, some of which are clearly non-sensory cells from glandular tissue, co-clustered with other cell types as well as a possibility that Obp2a is expressed below the detection level of our assay in neurons, which will require further experiments. We do not have any possible reason to confidently assign this cluster as a neuronal cell type, hence, we excluded it in downstream analysis of neurons.

We used the data from Hills et al., to compare co-expression characteristic of V2Rs, which is added as Figure 3-figure supplement 3.

(2) I found the UMAPs for the neurons somewhat difficult to interpret. Unlike Katreddi et al. 2022 or Hills et al. 2024, it's tricky to follow the developmental trajectories of the cells in the UMAP space. Perhaps the authors could try re-embedding the data using gene sets that don't include the receptors? It would also be interesting to see if the neuron clusters still cluster by receptor-type even when the receptors are excluded from the gene sets used for clustering. Plots relating the original clusters to the neuronal clusters, or dot plots showing marker gene expression for the neuronal clusters might both be useful. For example, right now it's difficult to interpret clusters like n8-13.

a) We have revised the UMAP in Figure 3A, and labeled mature, immature, progenitor neurons so that it is easier to follow the developmental trajectory.

b) In our revised text we have explicitly drawn equivalence between neuronal clusters from Figure 1 to re-clustered neurons in subsequent figures (Figure 3 and 4 in revised submission). For developmental analysis, we merged mature Gnao1, Gnai2 neuronal subclusters to two major clusters that are equivalent to original neuronal clusters in Figure 1. As UMAP is an arbitrary representation of cells, we also show expression of markers for major neuronal cell types in Figure 1C and Figure 3-figure supplement 1B, helpful in making the connection.

c) The purpose of re-clustering with higher resolution was to identify sub-populations within Gnao1 and Gnai1 neurons. It was useful to make sense of mature Gnao1 neurons, where family-C Vmn2r and H2-Mv expression maps onto distinct subclusters. Along with neuronal subclusters in revised Figure 3-figure supplement-1 we include a dot plot of gene expression markers.

d) In Figure 3-figure supplement-2, we show a comparison of neuronal clusters with and without VRs. Exclusion of VRs did not substantially alter mature neuron dichotomy into Gnao1/Gnai2. Only Gnao1 subclusters n1/n3 whose organization is dependent on family-C Vmn2r expression were affected, as well as redistribution of subcluster n8 from Gnai2 neurons. VR expression does not seem to be the primary determinant of VSN cluster identity.

**Reviewer #2 (Public Review):**
Summary:The study focuses on the vomeronasal organ, the peripheral chemosensory organ of the accessory olfactory system, by employing single-cell transcriptomics. The author analyzed the mouse vomeronasal organ, identifying diverse cell types through their unique gene expression patterns. Developmental gene expression analysis revealed that two classes of sensory neurons diverge in their maturation from common progenitors, marked by specific transient and persistent transcription factors. A comparative study between major neuronal subtypes, which differ in their G-protein sensory receptor families and G-protein subunits (Gnai2 and Gnao1, respectively), highlighted a higher expression of endoplasmic reticulum (ER) associated genes in Gnao1 neurons. Moreover, distinct differences in ER content and ultrastructure suggest some intriguing roles of ER in Gnao1-positive vomeronasal neurons. This work is likely to provide useful data for the community and is conceptually novel with the unique role of ER in a subset of vomeronasal neurons. This reviewer has some minor concerns and some suggestions to improve the manuscript.Strengths:(1) The study identified diverse cell types based on unique gene expression patterns, using single-cell transcriptomic.(2) The analysis suggests that two classes of sensory neurons diverge during maturation from common progenitors, characterized by specific transient and persistent transcription factors.(3) A comparative study highlighted differences in Gnai2- and Gnao1-positive sensory neurons.(4) Higher expression of endoplasmic reticulum (ER) associated genes in Gnao1 neurons.(5) Distinct differences in ER content and ultrastructure suggest unique roles of ER in Gnao1-positive vomeronasal neurons.(6) The research provides conceptually novel on the unique role of ER in a subset of vomeronasal neurons, offering valuable insights to the community.Weaknesses:(1) The connection between observations from sc RNA-seq and EM is unclear.(2) The lack of quantification for the ER phenotype is a concern.

We have extensively quantified the ER phenotype as shown in Figure 7, Figure 7-figure supplement-1 in our revised version. We would like to point out that the connection between scRNA-seq and EM was made due to our observations in the same figures, that levels of a number of ER luminal and ER membrane proteins were higher in Gnao1 compared to Gnai2 neurons. This led us to hypothesize a differential ER content or ultrastructure, which was verified by EM.

**Reviewer #3 (Public Review):**
Summary:In this manuscript, Devakinandan and colleagues have undertaken a thorough characterization of the cell types of the mouse vomeronasal organ, focusing on the vomeronasal sensory neurons (VSNs). VSNs are known to arise from a common pool of progenitors that differentiate into two distinct populations characterized by the expression of either the G protein subunit Gnao1 or Gnai2. Using single-cell RNA sequencing followed by unsupervised clustering of the transcriptome data, the authors identified three Gnai2+ VSN subtypes and a single Gnao1+ VSN type. To study VSN developmental trajectories, Devakinandan and colleagues took advantage of the constant renewal of the neuronal VSN pool, which allowed them to harvest all maturation states. All neurons were re-clustered and a pseudotime analysis was performed. The analysis revealed the emergence of two pools of Gap43+ clusters from a common lineage, which differentiate into many subclusters of mature Gnao1+ and Gnai2+ VSNs. By comparing the transcriptomes of these two pools of immature VSNs, the authors identified a number of differentially expressed transcription factors in addition to known markers. Next, by comparing the transcriptomes of mature Gnao1+ and Gnai2+ VSNs, the authors report the enrichment of ER-related genes in Gnao1+ VSNs. Using electron microscopy, they found that this enrichment was associated with specific ER morphology in Gnao1+ neurons. Finally, the authors characterized chemosensory receptor expression and coexpression (as well as H2-Mv proteins) in mature VSNs, which recapitulated known patterns.Strengths:The data presented here provide new and interesting perspectives on the distinguishing features between Gnao1+ and Gnai2+ VSNs. These features include newly identified markers, such as transcription factors, as well as an unsuspected ER-related peculiarity in Gnao1+ neurons, consisting of a hypertrophic ER and an enrichment in ER-related genes. In addition, the authors provide a comprehensive picture of specific co-expression patterns of V2R chemoreceptors and H2-Mv genes.Importantly, the authors provide a browser (scVNOexplorer) for anyone to explore the data, including gene expression and co-expression, number and proportion of cells, with a variety of graphical tools (violin plots, feature plots, dot plots, ...).Weaknesses:The study still requires refined analyses of the data and rigorous quantification to support the main claims.The method description for filtering and clustering single-cell RNA-sequencing data is incomplete. The Seurat package has many available pipelines for single-cell RNA-seq analysis, with a significant impact on the output data. How did the authors pre-process and normalize the data? Was the pipeline used with default settings? What batch correction method was applied to the data to mitigate possible sampling or technical effects? Moreover, the authors do not describe how cell and gene filtering was performed. The data in Figure 7-Supplement 3 show that one-sixth of the V1Rs do not express any chemoreceptor, while over a hundred cells express more than one chemoreceptor. Do these cells have unusually high or low numbers of genes or counts? To exclude the possibility of a technical artifact in these observations, the authors should describe how they dealt with putative doublet cells or debris. Surprisingly, some clusters are characterized by the expression of specific chemoreceptors (VRs). Have these been used for clustering? If so, clustering should be repeated after excluding these receptors.The identification of the VSN types should be consistent across the different analyses and validated. The data presented in Figure 1 lists four mature VSN types, whereas the re-clustering of neurons presented in Figure 3 leads to a different subdivision. At present, it remains unclear whether these clusters reflect the biology of the system or are due to over-clustering of the data, and therefore correspond to either noise or arbitrary splitting of continua. Clusters should be merged if they do not correspond to discrete categories of cells, and correspondence should be established between the different clustering analyses. To validate the detected clusters as cell types, markers characteristic of each of these populations can be evaluated by ISH or IHC.There is a lack of quantification of imaging data, which provides little support for the ERrelated main claim. Quantification of co-expression and statistics on labeling intensity or coverage would greatly strengthen the conclusions and the title of the paper.

a) scRNA-seq data analysis methods: Our revised submission has expanded on the methods section with details of parameters, filtering criterion and software used.

b) Inclusion/exclusion of VRs: Figure 3-Figure supplement-2 of our revised submission shows a comparison of neuronal sub-clusters with and without VRs. Overall sub-cluster identities were not affected by VR exclusion, except for Gnao1 sub-clusters n1/n3 -governed by family C Vmn2r1/Vmn2r2 and redistribution of Gnai2 cluster n8. The minimal effect of VRs on Gnai2 sub-clustering can also be confirmed by lack of V1R in the dot plot showing markers of neuronal clusters.

c) Neuronal clusters and potential over-clustering: we pooled neuronal cells from Figure-1 and re-clustered to identify sub-populations within Gnao1 and Gnai1 neurons. Several neuronal sub-clusters identified by us including progenitors, immature neurons and mature neurons are validated by previous studies with wellknown markers. Amongst the mature neurons, the biological basis of four Gnao1 neuron sub-clusters (n1-n4) is discussed in our co-expression section (Figure 4AE) and these are also validated by previous experimental studies. These Gnao1 clusters are organized according to the expression of family-C V2Rs (Vmn2r1 or Vmn2r2) as well as H2M_v_ genes. Within Gnai2 sub-clusters, n12 and n13 exclusively express markers that distinguish them from n8-n11 which we have described in our revised version. However, n8-n11 do not have definitive markers and whether these sub-clusters are part of a continuum or over-clustered, will require further extensive experiments and analysis. We prefer to show all subclusters, including Gnai2 sub-clusters, in Figure 3-Figure supplement-1, along with a dot plot of sub-cluster gene expression, so that this data is available for future experiments and analysis. We share the concern that some Gnai2 sub-clusters may not have an obvious biological basis at this time. Hence in our revised submission, we have merged mature Gnao1 and mature Gnai2 sub-clusters for the developmental analysis shown in Figure 3A.

d) Quantification of the ER phenotype: In our revised submission, we provide extensive quantification of the ER phenotype in Figure 7, Figure7-figure supplement-1.

e) We think that the cells expressing zero as well as two V1Rs are real and cannot be attributed to debris or doublets for the following reasons:

i) Cells expressing no V1Rs are not necessarily debris because they express other neuronal markers at the same level as cells that express one or two V1Rs. For instance, Gnai2 expression level across cells expressing 0, 1, 2 V1Rs is the same, which we have included in Figure 4-figure supplement 4-C of our revised submission. Higher expression threshold value used in our analysis may have somewhat increased the proportion of cells with zero V1Rs. Similarly, Gnao1 levels across cells expressing multiple V2Rs and H2-M_v_ per cell stay the same, indicating that these are unlikely to be doublets (Figure 4 I-K). The frequency of each co-expression combination (Supplementary Table 7 and 8) itself is an indication of whether it is represented by a single cell or an artifact.

ii) Cells co-expressing V1R genes: We listed the frequency of cells co-expressing V1R gene combinations in Supplementary table - 8. Among 134 cells that express two V1Rs, 44 cells express Vmn1r85+Vmn1r86, 21 express Vmn1r184+Vmn1r185, 13 express Vmn1r56+Vmn1r57, 6 express Vmn1r168+Vmn1r177, and so on. Doublets generally are a random combination of two cells. Here, each specific co-expression combination represents multiple cells and is highly unlikely by random chance. Some of the co-expression combinations we reported were also identified and verified experimentally in Lee et al., 2019 and Hills et. al., 2024.

**Recommendations for the authors:**

**Reviewing Editor (Recommendations for the Authors):**
The editor had a query about the analysis of FPRs, which are a third family of sensory receptors in the rodent VNO.

FPRs were found in our study as expressed in subsets of Gnai2 and Gnao1 neurons as well as non-neuronal cells. These can be easily searched in www.scvnoexplorer.com. For instance, Fpr1 and Fpr2 are expressed in immune cell clusters - 2,6,8,10; whereas Fpr-3 is expressed in Gnao1 subcluster n1. Consistent with earlier reports (10.1073/pnas.0904464106, 10.1038/nature08029) expression of Fpr-rs3, Fpr-rs4, Fprrs6, Fpr-rs7 is restricted to Gnai2 neurons, of which Fpr-rs3 and Fpr-rs4 are limited to Tmbim1+ Gnai2 neurons.

**Reviewer #1 (Recommendations For The Authors):**
(1) The reference to "genders" on page 3 should be changed to "sexes".

We have modified the text.

(2) Did the authors identify any Ascl1+ GBCs in their data?

Ascl1+ GBCs were identified and are now marked in our revised version Figure3-figure supplement 1B.

(3) The plots in Figures 1B and 2B say they're depicting gene "Expression", but it looks like the gene expression was z-scored. If so, the authors should describe how the expression was scaled.

We have modified the legend title to ‘scaled expression’ and described the basis of scaling in the methods section of our revised version.

(4) The main text mentions Figure 2C, but maybe this refers to the right part of Figure 2B?

Panel 2C was mistakenly not marked in the figure. We have now marked it in revised Figure 2.

(5) The authors should attempt to describe the other branch points in the trajectory shown in Figure 3A. If they don't seem biologically plausible, then the authors might want to reconsider using Slingshot for their analyses.

We do not seek to claim additional branch points within mature Gnao1 or Gnai2 neurons from our analysis. Whether there exist additional branch points leading to subcategories within mature neurons, requires extensive experimental investigation. Hence, in our revised submission, we have merged mature Gnai2 / Gnao1 subclusters for pseudotime developmental analysis and to keep our analysis focused on the single branch point at immature neurons.

(6) The most significantly enriched gene in Figure 3B in immature Gnao1+ neurons is Cnpy1, which is also an ER protein. It could also be interesting to look at its expression or speculate on its function in immature neurons.

Multiple ER genes were found to be enriched in Gnao1 neurons. We would not be comfortable speculating on the function of individual genes, without a proper study, which is beyond the scope of this manuscript.

(7) For figures with pseudo-colored expressions, it would be useful to have color bars. I'm also not sure the pseudocolors are necessary; presenting the data in grayscale or a single color like green might also be sufficient.

We used pseudocolor in the IHC images of ER proteins, because there is a wide variation in the fluorescence signal intensity across apical to basal axis for various proteins. In some cases, gray scale images could lead to the false impression that there is no signal in apical Gnai2 neurons, whereas pseudocolor shows low fluorescence level in these neurons. We have added intensity scale bar to the figures in our revision version.

(8) For in situ images with two colors it would be more colorblind-friendly to use green and magenta rather than green and red.

Since no single color palette can help readers with different types of colorblindness, we decided to rely on user’s operating systems that offer rendering of the images to a color palette based on their type of colorblindness. We believe this would be a better option as described here: https://markusmeister.com/2021/07/26/figure-design-for-colorblindreaders-is-outdated/

(9) The heatmap in Figure 7E would likely look more accurate without interpolation/aliasing/smoothing.

We have not performed smoothening on any of the heatmaps. We have noticed that sometimes heatmaps take time to load in software (such as Adobe Acrobat) leading to the impression of smoothing. Changing the zoom level or reopening the file may fix this.

(10) Rather than just citing the literature on the unfolded protein response in the MOE, it could be useful to cite work on the ATF5 expression and the UPR in the VNO e.g.

10.1101/239830v1 or 10.12688/f1000research.13659.1.

We have cited and commented on the ATF5 VNO expression in our discussion.

(11) I might try to condense the discussion. Additionally, in the discussion, the section on receptor co-expression comes before that on the VNO ER, so I might consider reorganizing the figures and results to present all of the scRNA-seq analyses (including the receptor co-expression figure) first before the figures on the ER.

We welcome this suggestion and have reorganized figures and results such that the scRNA-seq analysis flow is maintained before ER results.

**Reviewer #2 (Recommendations For The Authors):**
(1) Upregulation of ER-related mRNAs and expanded ER lumen in Gnao1-positive neurons is interesting, but the connection between these observations is unclear. The authors can strengthen the link by adding immunohistochemistry of representative ER proteins to test if the upregulation of mRNAs related to ER results in increased levels of these proteins in the ER of these neurons.

Connection between scRNA-seq and EM was made due to our observations that levels of a number of ER luminal and membrane proteins were higher in Gnao1 compared to Gnai2 neurons (Figure 7, Figure 7-figure supplement-1 in our revised submission). This led us to hypothesize a differential ER content or ultrastructure, which was verified by EM. We have also addressed the question of whether upregulation of mRNAs related to ER proteins results in their increased levels (Figure 7-figure supplement-2). In some cases, for example Hspa5 (Bip), mRNA as well as protein levels are upregulated in Gnao1 neurons (see Figure 3A volcano plot, Figure 5-figure supplement-1 RNA-ISH, Figure 7-figure supplement-1 comparison of mRNA levels, Figure 7F immunofluorescence). However, there are other genes in the same figures, for which mRNA levels are not upregulated, yet protein levels are higher in Gnao1 neurons. As mentioned in our text and discussion, upregulated mRNA levels as well as post-transcriptional mechanisms are both likely to play a role in upregulating ER protein levels in Gnao1 neurons.

(2) In Figure 3, the authors seemed to exclude cluster 13 from Figure1 in the pseudotime analysis without justification.

Cluster13 has markers such as Obp2a, Obp2b, Lcn3. We confirmed via RNA-ISH (Figure 2-figure supplement-1A in our revised submission) that Obp2a maps to cells from glandular tissue on the non-sensory side. Cluster-13 also has cells expressing Vmn1r37, which typically is expressed in neuronal cells. However, we do not see Obp2a mRNA in the sensory epithelium. It is possible that cluster-13 comprises a heterogenous mixture of cells, some of which are non-sensory glandular cells, co-clustered with other cell types as well as the possibility that Obp2a is expressed in neurons, below the detection level of our assay. Further experiments will be required to distinguish between these possibilities. We do not have any possible reason to confidently assign this cluster as a neuronal cell type, hence, it was excluded in the downstream analysis of neurons.

(3) In Figure 3, the line appears to suggest that Gnao1-positive cells can be progenitors of Gnai2-positive cells. Please clarify.

We thank the reviewer for pointing this out. We did not seek to give the impression that Gnao1 cells can be progenitors of Gnai2 cells. This may be due to the placement of dots in the trajectory leading to misinterpretation and the UMAP itself. We have modified the pseudotime trajectory in our revised version to make it more intuitive.

(4) Figure 3: Please label pseudotime lineage cluster identities.

Cluster identities are now labeled in Figure 3A pseudotime lineage as well as in Figure 3-figure supplement-1 dot plot.

(5) Figure 4: Please label the genes used for in situ hybridization in the volcano plot.

Genes used for RNA-ISH are labeled (bold font) in the volcano plot in Figure 5A.

(6) Figure 4: Please clarify which genes shown in the in situ hybridization figures correspond to which GO terms.

We have added supplementary table-10 containing gene ontology terms associated with genes for which RNA-ISH was performed.

(7) The EM shown in Figure 5 makes this work unique and intriguing. However, the lack of quantification for the ER phenotype is a concern. For example, does the ER area of a given cell correlate with the relative position of the cells along the apical-basal axis of the vomeronasal organ? What about the ER morphology in the progenitor cells?

We show here a quantification of the ER area from the low magnification EM image shown in Figure 8A. The ER area shows an increase going towards the basal side of the cross-section. However, this quantification is complicated by the following factors: (a) Processing for EM, results in some shrinkage of the tissue, (b) Gnao1 neurons follow an invaginating pattern in cross-sections. Due to these reasons, some Gnao1 neurons could come very close to, and at times lie adjacent to Gnai2 neurons in EM cross-section. Due to a lack of contrast, it is harder to identify the ER within the cell at low mag, especially in the apical zone. The plot shown here does indicate that roughly, the ER area of a cell correlates with its position along the apical-basal axis. In our revised submission, we have quantified the fluorescence intensities of various ER proteins along the apical basal axis from confocal images (Figure 7, Figure 7-figure supplement-1).

**Author response image 2. sa4fig2:** ROIs (yellow) are manually drawn in the sensory epithelium, wherever possible to identify ER without ambiguity. Area and centroid of ROI are calculated and x coordinates of centroid of each ROI are used to position ER area along the apical-basal axis as shown in the plot below.

Establishing ER ultrastructure in progenitor or immature cells, as well as unambiguous quantification of ER area in mature neurons, requires identification of these cells in crosssections using fluorescent molecular markers, followed by performing correlative light and electron microscopy (CLEM). This procedure being technically challenging is beyond the scope of our manuscript.

**Reviewer #3 (Recommendations For The Authors):**
(1) The main claim is about ER differences between Gnao1+ and Gnai2+ VSN. The ISH, IHC, and EM microscopy images are not quantified and, therefore, poorly support this main claim.

In our revised submission, we provide extensive quantification of the ER phenotype in Figure 7, Figure7-Figure supplement-1. Quantification of ER area from EM images is challenging and described above it in our response to reviewer #2 recommendation 7.

(2) The annotation of VSN subclusters should be more rigorous, consistent throughout the paper (VSN clusters are inconsistent between Figure 1 and Figure 3, and the multiplication of subclusters in Figure 3 is not discussed), and verified (using ISH or IHC) that they reflect discrete, actual cell types. The authors should provide a list of differentiating marker genes for the clusters in Figure 3. At present, it remains unclear whether these clusters are the result of over-clustering of cells (and therefore represent either noise or arbitrary splits of continua) or whether they reflect the biology of the system. Subsequent characterization of these curated VSN subtypes (as done in Figure 4) would add value to the study.

We pooled neuronal cells from Figure-1 and re-clustered at higher resolution to identify subtypes. Several neuronal sub-clusters identified by us including progenitors, immature neurons and mature neurons are validated by previous studies with well-known markers. Amongst the mature neurons, the biological basis of four Gnao1 neuron sub-clusters (n1n4) is discussed in our analysis and these are also validated by previous experimental studies. These Gnao1 clusters are organized according to the expression of family-C V2Rs (Vmn2r1 or Vmn2r2) as well as H2Mv genes. Within Gnai2 sub-clusters, n12 and n13 exclusively express markers that distinguish them from n8-n11 which we have described in our revised version. However, Gnai2 n8-n11 do not have definitive markers and whether these sub-clusters are part of a continuum or over-clustered, will require further extensive experiments and analysis. We prefer to show all sub-clusters, including Gnai2 sub-clusters, in Figure 3-Figure supplement-1, along with a dot plot of sub-cluster gene expression, so that this data is available for future experiments and analysis. We share the concern that some Gnai2 sub-clusters may not have an obvious biological basis at this time. Hence in our revised submission, we have merged mature Gnao1 and mature Gnai2 sub-clusters for the developmental analysis shown in Figure 3A.

(3) Some clusters are characterized by the expression of specific chemoreceptors (VRs). Have these been used for clustering? If so, clustering should be repeated after excluding these receptors.

Figure 3-Figure supplement-2 of our revised submission shows a comparison of neuron clusters with and without VRs. We also describe in the results, specific clusters that are affected by exclusion of VRs.

(4) Given the title and the data, the paper should be structured around its main claim (i.e. differential ER environment between VSN types). For example, Figure 7, which deals with the characterization of receptor expression and co-expression in VSNs, is sandwiched between the validation of ER substructure (Figure 6) and the timing of coexpression of ER chaperone genes (Figure 8). The data presented in Figure 7 would fit better if used as a validation of the dataset prior to the investigation presented in the current Figure 4. In addition, we suggest that expression and co-expression diagnostics should be used to filter cells for subsequent analyses.

We appreciate this suggestion and have reorganized the figures in our revised version. Our subsequent analysis showing enrichment of ER related genes at RNA, protein level covers all Gnao1 neurons and is not restricted to a specific subset. This is reflected in the ISH and IHC of ER genes.

(5) Figure 7-Supplement 3 suggests the presence of co-expressed V1Rs in VSNs. It is unclear from the data presented whether these co-expressing cells are artifactual cell doublets and should be removed from the analysis or whether the expression of the coexpressed receptors reflects a reality. To better address this observation, one may want to see the expression levels of the individual co-expressed V1rs in Figure 7-Supplemet 3 rather than the sum of V1r expression. I am also concerned about the unusually high frequency of "empty" neurons (i.e. without expressed VRs). Could these be debris?

We think that the cells expressing zero as well as two V1Rs are real and cannot be attributed to debris or doublets for the following reasons:

i) Cells expressing no V1Rs are not necessarily debris because they express other neuronal markers at the same level as cells that express one or two V1Rs. For instance Gnai2 expression level across cells expressing 0, 1, 2 V1Rs is the same, which we have included in Figure 4-figure supplement 4-C of our revised submission. Higher expression threshold values used in our analysis may have somewhat increased the proportion of cells with zero V1Rs. Similarly, Gnao1 levels across cells expressing multiple V2Rs and H2-M_v_ per cell stay the same, indicating that these are unlikely to be doublets (Figure 4 I-K). As doublets are formed randomly, the frequency of each co-expression combination (Supplementary Table 7 and 8) itself is an indication of whether it is represented by a single cell or an artifact.

ii) Cells co-expressing V1R genes: All cells used for co-expression analysis were filtered via an expression threshold (Figure 4-figure supplement 1D), which eliminates cells with low counts of V1R expression. We listed the frequency of cells co-expressing V1R gene combinations in Supplementary table - 8. Among 134 cells that express two V1Rs, 44 cells express Vmn1r85+Vmn1r86, 21 express Vmn1r184+Vmn1r185, 13 express Vmn1r56+Vmn1r57, 6 express Vmn1r168+Vmn1r177, and so on. Doublets generally are a random combination of two cells. Here, each specific co-expression combination represents multiple cells and is highly unlikely by random chance. iii) Some of the co-expression combinations we reported were identified earlier and verified experimentally in Lee et al., 2019 using FACS based single collection in 96-well plates following the cellseq-2 protocol with very low chance of doublets, and Hills et. al., 2024.

(6) The authors use either dot plots or scatter plots to show gene expression in cell clusters. It looks nice, but it is very difficult to deduce population levels of expression from these plots. Could we see the distribution of gene expression across clusters using more quantitative visualizations such as violin or box plots?

Dot plots are majorly used in our manuscript to show markers of cell clusters in Figure 1, Figure 2 and Figure 3-figure supplement 1. We would like to show at least 5 gene markers for each cluster that are important to identify the cell type. Using violin plot or bar plot for this will make the panel extremely big and overwhelming, especially with 16 clusters in Figure 1 and 13 clusters in Figure 3-figure supplement 1 or make the bars/violin too small to interpret. Hence, for the sake of simplicity, we used dot plots to give our reader a birds-eye of gene expression differences across clusters. Scatter plots were used when we want to compare the expression levels of genes between male and female samples and show the expression of two genes (VRs) simultaneously in a single cell. This cannot be achieved by Violin/box plot. However, we have made our dataset available at scvnoexplorer.com to explore the expression patterns across cell clusters with different visualization options, including violin or box plots.

(7) To investigate whether sex might bias clustering, the authors calculated the Pearson coefficient of gene expression between sexes for each cluster. Given the high coefficient observed across all clusters (although no threshold is used), the authors conclude that there was no bias. While the overall effect may show a strong similarity in gene expression in each cluster between the sexes, this overlooks all the genes that are significantly differentially expressed. It would be worth investigating and discussing these differences. Relatedly, what batch correction method was applied to the data (to mitigate any possible sampling or technical effect)?

We chose the Pearson coefficient as a representative parameter to show that there is no bias. In addition, we have performed differential expression analysis for each cluster and the results are in supplementary table-1. Except known sexually dimorphic genes, other genes are not differentially expressed significantly with adjusted p-values greater than 0.05. This was also shown by earlier studies using bulk RNAseq (doi.org/10.1371/journal.pgen.1004593, doi.org/10.1186/s12864-017-4364-4). We used depth normalization to integrate samples and described this in the methods section of our revised version.

(8) We found the method description to be incomplete for the single-cell RNA sequencing analyses. The method section should include a detailed explanation of the code used by the authors to analyze the data. The Seurat package has many available pipelines for single-cell RNA-seq analysis, which have a major impact on the output data. It is therefore imperative to describe which of these pipelines were used and whether the pipeline was run with default settings.

Our revised submission has expanded on the methods section with details of parameters, filtering criterion and software used.